# Add drop multiplexers for terahertz communications using two-wire waveguide-based plasmonic circuits

Yang Cao ⬚ [1], Kathirvel Nallappan[1], Guofu Xu[1] & Maksim Skorobogatiy[1]✉

Terahertz (THz) band is considered to be the next frontier in wireless communications. The emerging THz multiplexing techniques are expected to dramatically increase the information capacity of THz communications far beyond a single channel limit. In this work, we explore the THz frequency-division multiplexing modality enabled by an add-drop multiplexer (ADM) design. Based on modular two-wire plasmonic waveguides fabricated using additive manufacturing and metallization techniques, we demonstrate four-port THz ADMs containing grating-loaded side couplers for operation at ~140 GHz carrier frequency. Particular attention is paid to the design of plasmonic waveguide Bragg gratings and directional couplers capable of splitting broadband THz light into spectral and spatial domains. Finally, we demonstrate multi/demultiplexing of THz signals with bit rates up to 6 Gbps using the developed ADMs. We believe that the proposed plasmonic circuits hold strong potential to provide robust integrated solutions for analog signal processing in the upcoming THz communications.

[1] Department of Engineering Physics, École Polytechnique de Montréal, Montreal, QC H3T 1J4, Canada. ✉email: maksim.skorobogatiy@polymtl.ca

The global mobile data traffic is expected to reach 77.49 exabytes per month by the year 2022[1]. According to the Shannon-Hartley theorem[2], in the presence of noise, the maximal bit rate supported by the transmission channel is proportional to the available bandwidth. Currently, most of the wireless systems operate in the overcrowded microwave band which is not sufficient to meet the bandwidth demand of the near future. Shifting the carrier wave towards higher frequencies is inevitable to accommodate the imminent surge in data volume[3,4]. Therefore, the terahertz (THz) frequency band (0.1–10 THz) is considered as the next frontier in wireless communications[5,6]. To date, there are already several demonstrations of free-space ultra-high bit rate data transmission (>100 Gbps) using a single-channel THz link together with various optical multiplexing (ex. optical frequency and polarization division multiplexing) and modulation techniques (ex. quadrature amplitude modulation and quadrature phase-shift keying)[7–9]. The capacity of the THz communication system can be further increased by using various THz multiplexing techniques[10–12]. Among them, frequency-division multiplexing (FDM), in which several discrete carrier frequencies support distinct users, is routinely used in fiber optic and wireless communications to multiply the data-throughput capacity. In optical networks, one typically uses carrier waves in the C-band (infrared) with channel spacing <10 GHz, thus allowing simultaneous transmission of hundreds of channels with a hundred MHz bandwidth each[13]. Furthermore, in the microwave/RF domain, the FDM technology is also prevalent in the IEEE 802.11 wireless and mobile networks. Particularly, in 4G LTE and 5G cellular communication networks, the orthogonal frequency division multiple access is implemented using electronic circuits for FFT and IFFT, where thousands of tightly packed sub-carriers within tens of MHz bandwidth are spaced by tens of kHz[14]. As THz frequencies find themselves between infrared and microwave bands in the electromagnetic spectrum, the THz FDM communications can benefit from both the optical and electronic solutions. The lower THz band 100–300 GHz, in particular, can accommodate multiple channels with bandwidths from several GHz to several 10 s of GHz, thus allowing spectral allocation and simultaneous utilization of numerous high-bandwidth channels both within the THz fiber-based and THz wireless communication modalities.

A key component in designing FDM systems is the add-drop multiplexer (ADM). It allows multiple carrier frequencies to share the same frontend, thus reducing the overall size and complexity of the communication networks[15,16]. In optical FDM networks, the add-drop multiplexing is usually realized by the combination of mature optical elements such as ring resonator, phase-shifted grating, Bragg grating, arrayed waveguide grating, circulator, and coupler on the platform of silica-based planar lightwave circuits and fiber arrays[17–20]. However, due to the lack of trusted material platforms, universal standards for THz communication components, as well as the relative abundance of potential, while untested, design and fabrications routes, the ADMs capable of operation at THz frequencies are still in the early development. At the same time, most of the reported THz ADMs suffer from low efficiency when compared with the infrared and microwave components of similar configurations.

Using the directional coupling between two adjacent waveguides or the leaky wave of a single deformed waveguide, THz ADMs have been proposed based on parallel-plate waveguides and silicon substrates[16,21–25] (see Supplementary Note 1). Particularly, the silicon substrates hold great potential in developing feasible THz ADM designs, such as diplexer and bandpass filters[26], ring resonators[27], and topological insulators[28]. Nevertheless, Add/Drop frequency tunability is problematic, and the top-of-the-line expensive fab infrastructure is required to fabricate silicon-based circuits. In addition, a THz ADM prototype composed of a lossy Y-coupler and an external low-reflectivity Bragg grating was also fabricated using two-wire plasmonic waveguides[29,30]. However, most of the reported THz ADMs contain only three ports, while to implement both channel dropping and adding of THz carrier waves, cascaded ADMs or four-port ADMs have to be employed in the communication networks.

In what follows, we report the design, 3D fabrication, and characterization of the four-port THz ADMs for FDM communications. The uniqueness of our approach, which is developed specifically for the THz wave band, lies in the combination of the plasmonic structure designs, robust device fabrication/prototyping techniques, modular subcomponent integration approach, intrinsic circuit tunability, and cost-effectiveness. From the point of view of fabrication, the fab-less additive manufacturing technique makes integrated THz devices accessible to anyone with a 3D printer, thus democratizing the field of device fabrication for the upcoming THz communications. Moreover, additive manufacturing allows native fabrication of the 3D integrated freeform circuits, thus opening the prospects for the high-density out-of-plane photonic integration, which is difficult to achieve with other techniques (see Supplementary Note 1 for a more detailed discussion).

More specifically, in the core of our system are micro-encapsulated two-wire plasmonic waveguides fabricated using stereolithography (SLA) 3D printing and wet chemistry metal deposition techniques[29]. Particularly, we use relatively low-cost widely available SLA printers to realize the two-wire photonic circuits featuring metalized surfaces on the resin support (see Supplementary Note 2 for this waveguide design). This fabrication route makes such circuits highly suitable for rapid prototyping and mass production. Resultant two-wire waveguides feature some of the lowest transmission and bending losses, very low group velocity dispersion (GVD), and broadband operation compared to other THz waveguides[31,32].

On the component integration level, the methods developed in this work show a pathway to the highly reconfigurable, modular construction of complex terahertz circuits, where individual components can be easily added, replaced, or subtracted to modify the device functionality. In addition to the FDM demonstrated in this work, the developed waveguide circuits can be also used in a four-wire configuration to realize THz polarization-division multiplexing[33], thus paving a way for supporting several modulation modalities in a single device. Furthermore, two-wire waveguides can be efficiently butt-coupled to rectangular waveguides, which are currently prevalent in microwave and mm-wave communication devices[29]. This allows two-wire waveguide-based optical components developed within this work to be, in principle, seamlessly integrated into the upcoming THz communication networks by completely forgoing free-space optics and complicated optical alignment. Moreover, the frequency response of various components developed in this work (Bragg grating, directional coupler, ADM) can be readily tuned. This can be accomplished by dynamically changing the physical properties or the geometry of the plastic substrate and/or a metallic film using either thermal or mechanical activation. As the air-core two-wire waveguides feature a highly flexible structure and easily accessible modal fields, various elements can be inserted into the air gap to modify or tune the waveguide propagation properties. Moreover, optical and electrical tuning of such waveguides is also possible as was recently demonstrated via insertion of a graphene sheet between the two wires, which was then tuned via optical pumping or electrical gating[34–37]. Such structures promise solutions for the important problem of dynamic band allocation in the THz communication networks[38] (see Supplementary Note 3 for a more detailed discussion).

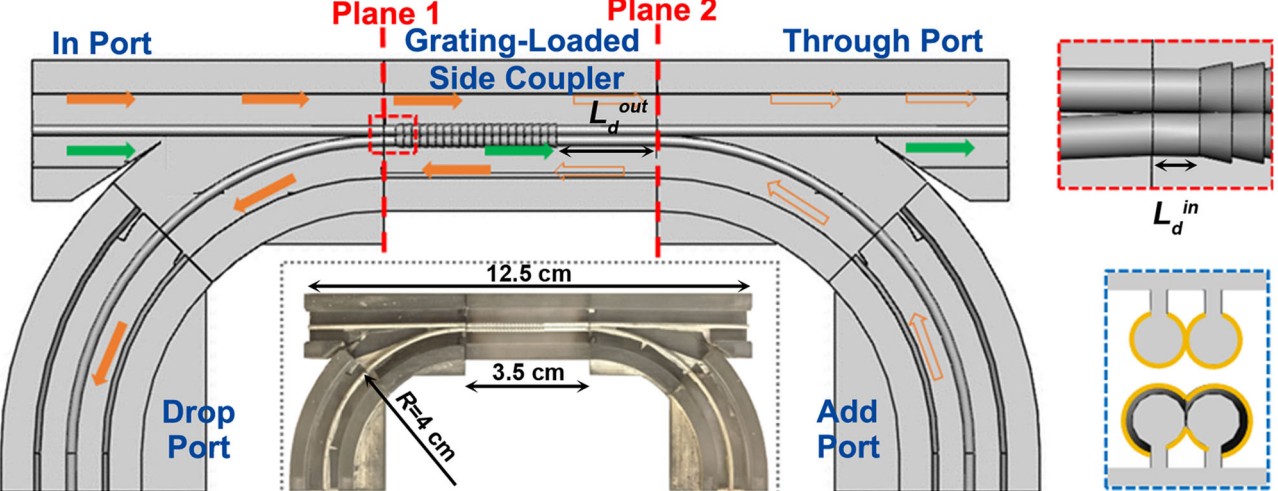

**Fig. 1 Schematic of the four-port ADM circuit comprising a grating-loaded side coupler placed between two Y-splitters.** $L_d^{in}$ and $L_d^{out}$ are the distances between the grating and the directional coupler end-facets (see an Insert in the red dotted box). The solid and hollow orange arrows indicate the paths of the dropped and added signals within the grating stopband. The green arrows indicate the path of a Through signal in the grating passband. Insert in the blue dotted box presents a cross-section of the grating-loaded side coupler. Insert in the black dotted box shows a fabricated four-port ADM circuit with the top half removed.

Finally, it is important to mention the versatile monolithic microwave integrated circuits (MMIC) that are built on planar semiconductor wafers, and whose application has extended into the THz field[39–41]. In comparison, while our plasmonic waveguide circuits are less compact than MMICs in the 2D plane, however, they carry several other advantages including high reconfigurability and tunability, ease of manufacturing and packaging, as well as the ability of native 3D integration. Furthermore, different from the MMICs whose operation requires external supply power, signal processing in the two-wire plasmonic circuits is completely passive and more reliable when operating in complex electromagnetic environments.

One key element of our ADM system is a waveguide Bragg grating (WBG) that serves as a wavelength-selective mirror[42]. By integrating the WBG into optical circuits, a tunable stopband and passband in the transmission and reflection spectra can be achieved. Hindered by the lack of THz circulators compatible with circuits[43,44], the directional coupler is used as an alternative diplexer to combine THz signals in the spatial domain, which is another key element of the proposed ADM circuit. Particularly, we developed an interferometric THz ADM where Bragg grating is inscribed on both side-coupled two-wire waveguides, similarly to the prior work in the mid-infrared range[17,45–47]. Compared with other ADM designs of similar configurations, such as grating-assisted contra-directional coupler[48] and grating-loaded Mach–Zehnder interferometer[49], the aforementioned design has advantages of compact structure, simple design criteria, single-mode operation, while allowing relatively relaxed fabrication tolerances.

In this work, first, we develop high-reflectivity WBGs based on the two-wire THz plasmonic waveguides, with one of the wires featuring a periodic array of end-to-end connected truncated cones. We experimentally demonstrate stopbands of up to 14 GHz with a center wavelength that can be positioned anywhere in the 120–160 GHz range. Second, we design and fabricate directional couplers based on two adjacent two-wire waveguides and then confirm experimentally that a several-centimeter-long coupler is capable of a full-cycle coupling at 140 GHz carrier frequency over a broad bandwidth exceeding 60 GHz. Finally, we demonstrate a four-port THz ADM circuit, where the grating-loaded side coupler allows spatially separating add and drop channels designed for the THz wavelengths that fall within the

grating stopband while letting all the other wavelengths within the coupler bandwidth pass through. Two wideband Y-splitters connected with the coupler are used to guide THz signals to/from the grating-loaded side coupler section from/to the desired ports of ADM circuits (see Fig. 1). Channel dropping and adding functionalities of ADMs with optimized Drop action and balanced Add/Drop action for several THz carrier waves in the vicinity of 140 GHz modulated up to 6 Gbps are demonstrated experimentally.

## Results

**Two-wire waveguide-based Bragg gratings.** The two-wire WBGs studied in this work are introduced by periodically varying the cross-section of a single wire inside a micro-encapsulated two-wire waveguide[29]. The choice of end-to-end connected truncated cones in the grating structure (see Fig. 2a) was experimentally found to be the most reliable and stable for printing compared to other alternative designs (see Supplementary Note 4 for details on the design and fabrication of the two-wire WBGs). Such WBGs typically feature high grating strengths and wide stopbands $\Delta\lambda \sim 0.1-0.2 \cdot \lambda_{Bragg}$ due to strong presence of the modal fields (confined in the gap between two wires) with the grating structure. The spectral position of the grating stopband can be estimated using standard quarter-wave condition $\lambda_{Bragg} \approx 2\Lambda n_{eff}$, where the effective refractive index of the grating mode for two-wire waveguides is close to that of air $n_{eff} \sim 1-1.1$.

As we will see in what follows, the operational bandwidth of ADM is determined by that of the WBGs used in its design. In this work, by comparing optical properties of WBGs we find the optimal ridge height of truncated cones to be $H = 0.21$ mm to ensure large operational bandwidth, reproducible optical performance, and manageable loss of ADMs. With the period length of a 20-periods WBG set to $\Lambda = 1.03$ mm, the resultant stopband center frequency is 140 GHz with the corresponding bandwidth of ~18 GHz (full width at half maximum (FWHM)) in numerical simulations.

We then compare the numerical and experimental results of the relative transmittance for the two-wire WBGs. We observe that a larger number of periods in a WBG results in more pronounced transmission dips (see Fig. 2b). We also find that the experimental WBGs show somewhat shallower transmission spectra within the grating stopbands compared to the numerical simulations. This is

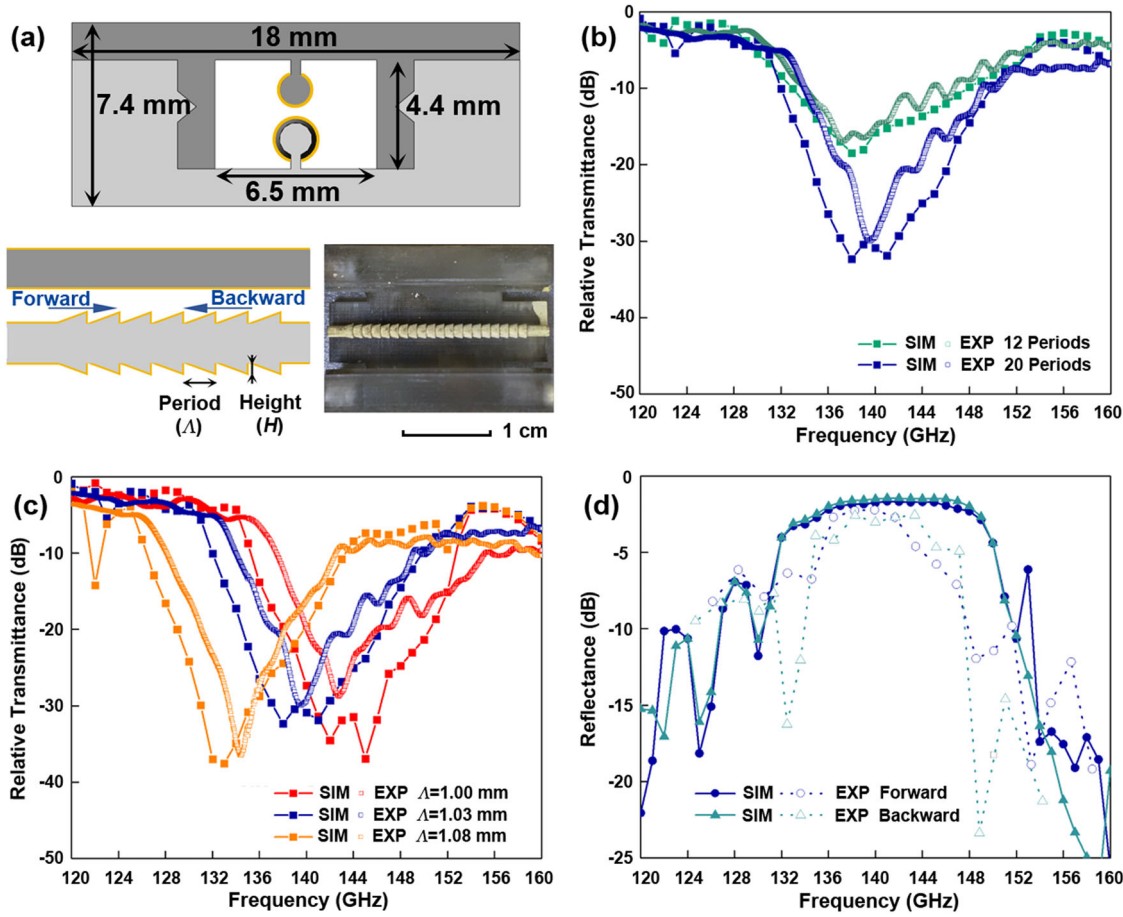

**Fig. 2 Two-wire waveguide Bragg grating. a** Schematic of the WBG cross-section, its side view, and a photograph of the fabricated WBG with the top half removed. The grating featuring a periodic sequence of truncated cones (light gray) and a cylindrical wire (dark gray) is suspended in the air on dielectric supports and encapsulated within a plastic cage. The numerical and experimental relative transmittance (by power) of the 2.5-cm-long WBG sections **b** comprising a different number of periods ($\Lambda = 1.03$ mm, $H = 0.21$ mm), and **c** containing 20 periods ($H = 0.21$ mm) as a function of the period length $\Lambda$. The THz light travels in the forward direction (see panel **a**). **d** The numerical and experimental reflectance (by power) of the 2.5-cm-long WBG sections comprising 20 periods ($\Lambda = 1.03$ mm, $H = 0.21$ mm) with two possible traveling directions of the THz light (see Supplementary Note 4 for more details).

mainly attributed to the field attenuation and scattering in the WBG structure. Nevertheless, with only 20 periods in the WBG structure a pronounced transmission dip near 140 GHz featuring a minimal transmittance of <1% is found experimentally within the grating stopbands of ~14 GHz (FWHM) bandwidth. Furthermore, we confirm the designability of the WBG spectral response by varying grating periodicity (see Fig. 2c). Reducing the grating period from 1.08 mm to 1 mm shifts the center frequency of the WBG stopband from 134 to 145 GHz, while measured WBG bandwidth remains largely unaffected. In addition, both the WBG transmittance and reflectance spectra are virtually symmetric when switching the traveling direction of the THz light despite the WBG nonsymmetric structure (see Supplementary Note 4). The WBG's high reflectance within the grating stopband was confirmed experimentally (see Fig. 2d), thus making them suitable for channel adding/dropping in the ADM circuits. Overall, we observe a good correspondence between the numerical and experimental spectral performance of the WBGs in terms of the center frequency, the bandwidth of the grating stopband, as well as minimal and maximal values of the normalized transmission and reflection coefficients.

**Two-wire waveguide-based directional coupler circuits.** Experimental realization of directional coupling using free-standing THz waveguides is typically challenging. This is due to the need for

precise alignment of such waveguides to maintain a fixed inter-waveguide separation (normally on a sub-mm scale), which neces-sitates the use of cumbersome holders, spacers, etc.[50–53]. In contrast, using high-definition 3D printing allows fabrication of the mono-lithic solutions that integrate waveguides, holders, and enclosure in a single step with precise control over the inter-waveguide separation. In the case of micro-encapsulated two-wire waveguides, the field of the fundamental mode is predominantly confined in the air gap between two wires[29]. Therefore, directional coupling between two separate waveguides is weak. To enhance the inter-waveguide cou-pling, we made the cylindrical wires belonging to two different two-wire waveguides to touch each other and increased the enclosure width (by a single wire diameter) to accommodate these two waveguides (see Fig. 3a, b).

The effective refractive indices ($n_e$ and $n_o$) of the even and odd supermodes supported by the side coupler are computed using the 2D mode solver tool of COMSOL Multiphysics, while the corresponding coupling length $L_c$ between the two modes is estimated as $L_c = \lambda / 2(n_e - n_o)$ (see Fig. 3c). For the presented design, the coupling length is $L_c \sim 24.5$ mm at ~140 GHz operational frequency, while even smaller coupling lengths (stronger coupling strengths) are possible to realize by allowing the two wires to partially overlap.

To spatially separate the THz light at the output end (Plane 2) and allow a two-port input (Plane 1), two Y-splitters were added

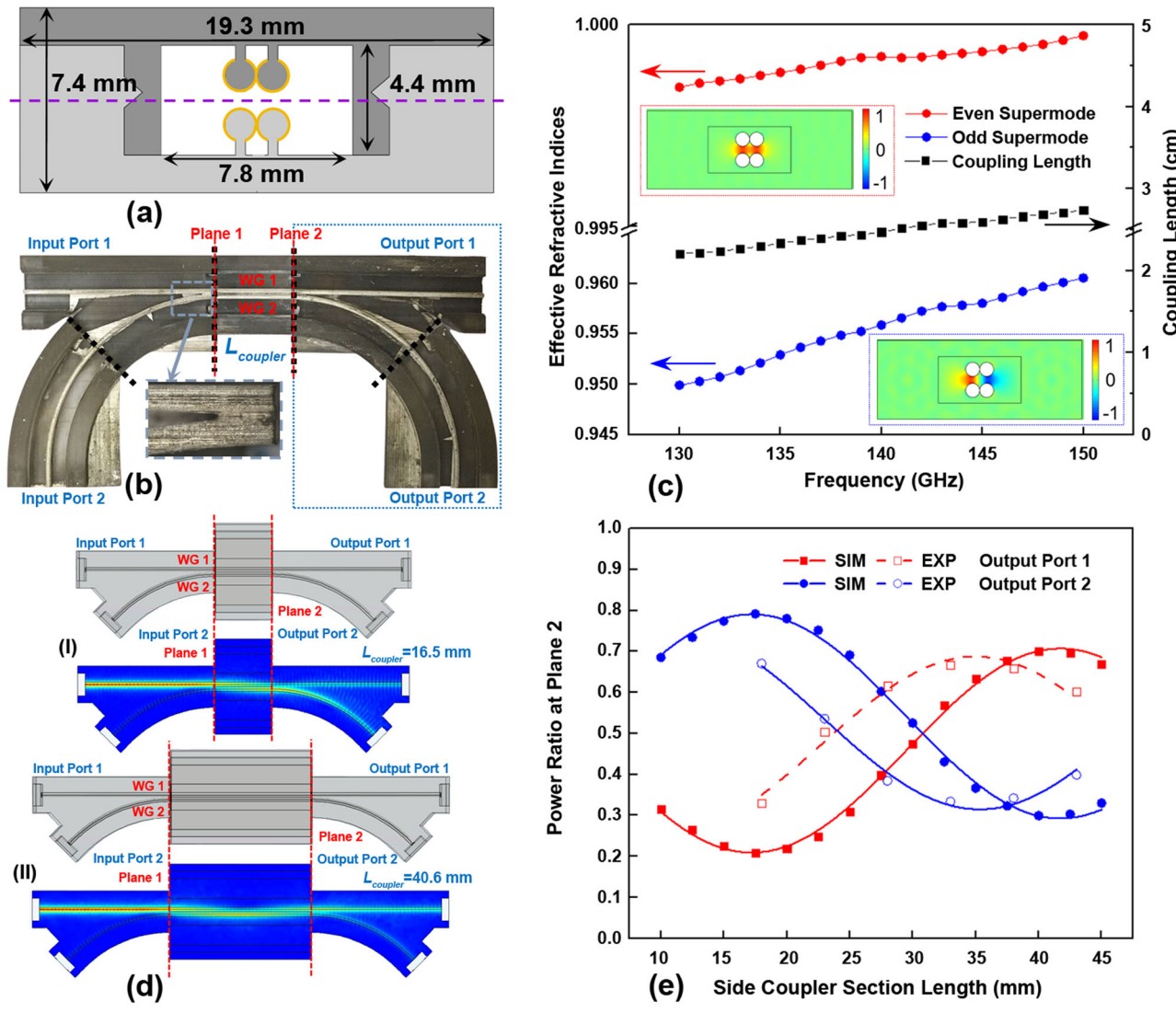

**Fig. 3 Two-wire waveguide-based directional coupler. a** Schematic of the coupler cross-section featuring two touching two-wire waveguides (WG1 and WG2) that are suspended in the air on dielectric supports and encapsulated within a plastic cage. **b** A photo of the fabricated directional coupler circuit (top half removed) with a side coupler section placed between two Y-splitters. The black dashed lines show interfaces between distinct modular sections of the experimental circuit. Insert: the enlarged view of the Y junction where curved and straight waveguides meet. **c** The numerical effective refractive indices of the odd and even supermodes guided by the directional coupler, and the corresponding coupling length. Inserts: electric field distributions (vertical component) for the even and odd supermodes. **d** 3D modeling of the directional coupler circuits, and the corresponding electric field distributions when using the fundamental mode of Input port 1 at 140 GHz as the excitation condition. (I) Case of the half-cycle coupler that results in the maximal power transfer from Waveguide 1 (WG1) to Waveguide 2 (WG2). (II) Case of the full-cycle coupler that results in the maximal power transfer from Waveguide 1 to Waveguide 2 and then back to Waveguide 1. **e** The power transmittance ratio for the Output ports 1 and 2 at Plane 2 as a function of the side coupler section length $L_{coupler}$.

at both ends of the side coupler element (see Fig. 3b). This configuration was then used to build the four-port ADMs. The Y-splitter features a curved two-wire waveguide gradually approaching a straight one in a wedge configuration (the two waveguides do not coalesce). The seamless connection between a Y-splitter and a side coupler section is assured as their cross-sections at the coupling planes (Planes 1 and 2) are the same by design. Therefore, butt-coupling of these components brings only a small insertion loss due to mismatch of the modal fields of straight and bent waveguides with otherwise identical cross-sections. We also note that Y-splitters used in the ADM have to be distinguished from the Y-couplers used for characterization of the WBG reflectance (see discussion in Supplementary Note 5), as the latter usually bring an additional 3 dB loss.

The optical performance of the complete directional coupler circuit as a function of the side coupler length (distance between Planes 1 and 2) was studied using a 3D model with ports of COMSOL Multiphysics. Particularly, we are interested in finding the relative signal intensities at the Output ports 1 and 2 (OP 1,2) of the directional coupler circuit containing side couplers featuring different lengths when assuming that all the power is launched into Input port 1 (IP 1). An immediate complication is that two branches of the Y-splitter used at the output side of the side coupler have significantly different losses as one of them is curved and longer compared to the straight one. Therefore, instead of simply plotting the power transmittance in the OP 1 and OP 2 assuming excitation at the IP 1 expressed via the corresponding elements of the scattering matrix ($T_{op\,1,2} = |S_{OP\,1,2;\,IP\,1}|^2$) (see Supplementary

Note 6), we characterize the performance of a side coupler section by estimating the ratio of power carried by the two waveguides (Waveguide 1 and Waveguide 2) at their end-facets (Plane 2) using,

$$T_{\text{CW}1,2} = |S_{\text{OP}1,2;\text{IP}1}|^2 / T_{Y-\text{branch}\,1,2} \tag{1}$$

$$\text{Ratio}_{\text{CW}1,2} = \frac{T_{\text{CW}1,2}}{T_{\text{CW}1} + T_{\text{CW}2}} \tag{2}$$

where $T_{Y\text{-branch}\,1,2}$ are numerically computed power transmittances of the two stand-alone branches of a Y-splitter (see Supplementary Note 5). The maximum power transfer of THz light at 140 GHz from Waveguide 1 to Waveguide 2 is achieved by a 16.5-mm-long coupler, while the power is cycled back into Waveguide 1 in a 40.6-mm-long coupler. The corresponding electric field distributions of the two directional coupler circuits within the plane marked by a violet dashed line in Fig. 3a are shown in Fig. 3d.

Compared to the beat length between the two supermodes ($2L_c \sim 49$ mm) which is computed theoretically using a 2D model solver, we find that the full-cycle coupling computed using a full 3D model is realized by the directional coupler circuit of a somewhat smaller length (40.6 mm) (see Fig. 3e). This difference mainly stems from the fact that coupling action between two waveguides persists over some distance in the Y-splitters as in this element the curved and straight waveguides run almost in parallel to each other over a certain distance. Finally, the incomplete power transfer in the coupler circuit is due to the significant loss difference of the two supermodes that have different field overlaps with the lossy resin cage (see Supplementary Note 6).

Experimental characterization of such circuits confirms sinusoidal behavior of the power coupling coefficients as a function of the side coupler length (see Fig. 3e). The experimentally found maximum power ratio for Waveguide 1 was obtained using a 35-mm-long side coupler section, which is somewhat shorter than the predicted one. This is related to the

deviation of the experimental geometry of Y-splitters from the ideal ones used in numerical modeling. As seen from the Insert of Fig. 3b, due to the limited resolution of a 3D printer, the two waveguides remain joint for several mm at the entrance of a Y-splitter, which results in stronger than expected inter-waveguide coupling in the fabricated Y-splitters.

Due to the contribution of Y-splitters to the coupling between the bus and dropping waveguides, we design the ADM circuit using a 35-mm-long side coupler section that enables the full-cycle coupling at ~140 GHz operational frequency with a broadband operation of over 60 GHz owing to the modest variation in the coupling length with frequency (see Supplementary Note 6).

**Grating-loaded side couplers for the ADM circuits.** The THz ADM circuits feature a grating-loaded side coupler placed between two Y-splitters. Particularly, the directional coupler circuit enabling full-cycle coupling is modified by inscribing Bragg gratings on a single set of two joint wires of the side coupler section (see Fig. 1). While propagating in the grating-loaded side coupler, the THz signal within the grating stopband that is launched into the In port on Waveguide 1 (Add port on Waveguide 2) is back-reflected by the gratings, and then transferred to Waveguide 2 (1) by the side coupler. The THz signal within the grating passband when launched into the In port propagates through the grating-loaded side coupler and into Through port (see Fig. 4a). Within the coupler bandwidth, the variation in the coupling length is moderate (see Fig. 3c), thus allowing high signal amplitude to be recorded at the Through port.

Separate numerical and experimental studies of Y-splitters (see Supplementary Note 5) reveal that in such structures THz light mostly propagates in the forward direction regardless of the

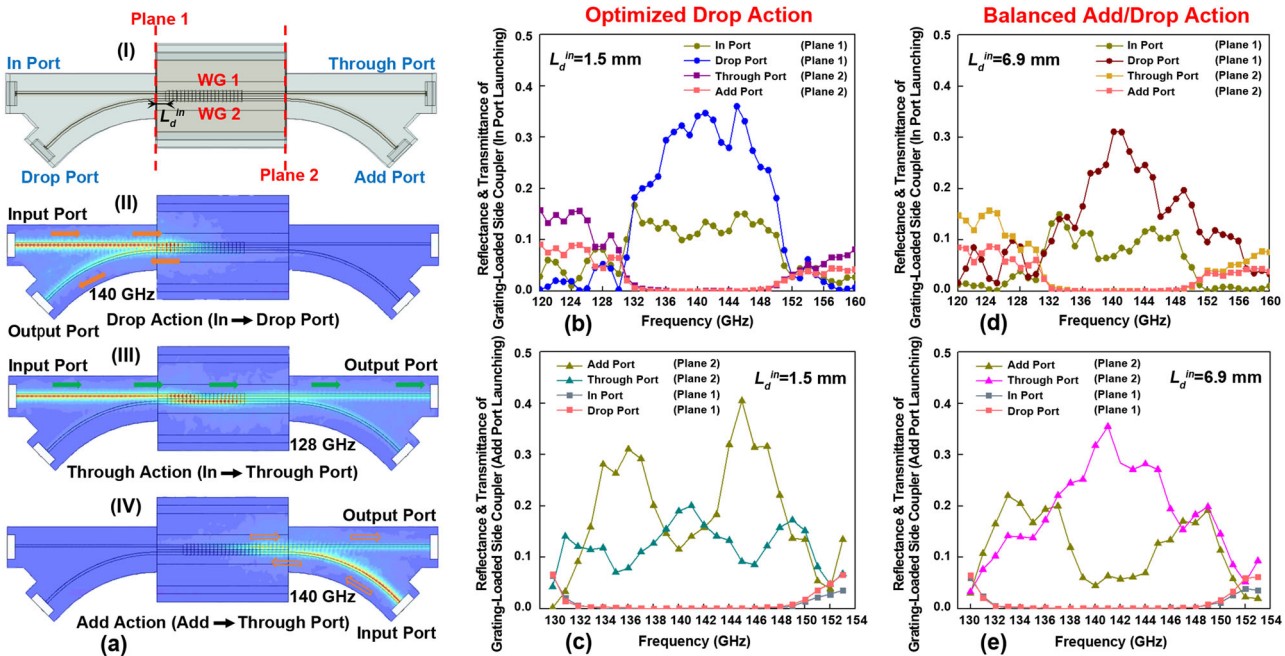

**Fig. 4 Numerical study of the grating-loaded side couplers. a** (I) 3D model of the ADM circuit used in simulations that comprises two Y-splitters and a 35-mm-long grating-loaded side coupler (20 grating periods with $\Lambda = 1.03$ mm, $H = 0.21$ mm). The computed electric field distributions in the ADM circuits for various launching conditions: (II) In port at 140 GHz (grating stopband), (III) In port at 128 GHz (grating passband), and (IV) Add port at 140 GHz (grating stopband). Optimized Drop Action: reflectance and transmittance of various ports under **b** In port launching and **c** Add port launching. Balanced Add/Drop Action: reflectance and transmittance of various ports under **d** In port launching and **e** Add port launching. Panels **b**–**e** are computed using Eq. (1) and data from (**a**).

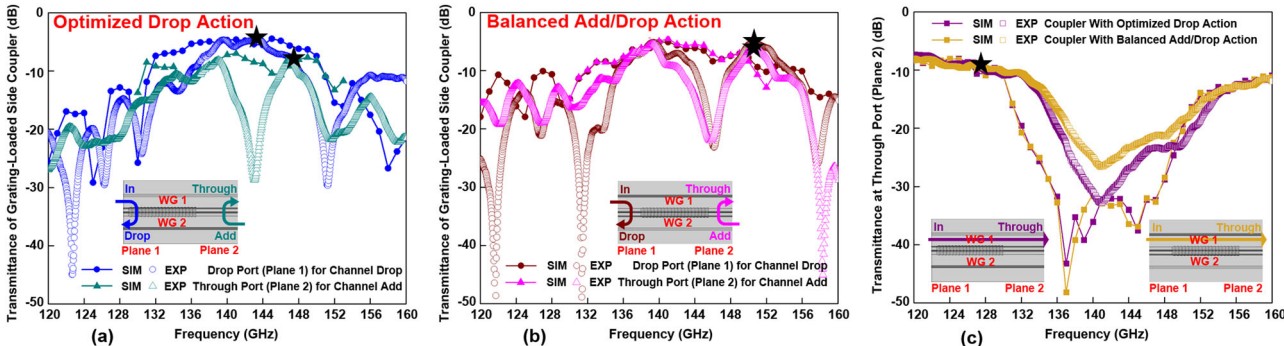

**Fig. 5 Optimized grating-loaded side couplers.** Power transmittances of the grating-loaded side couplers for **a** an ADM with the optimized Drop action ($L_d^{in}$ = 1.5 mm) and **b** an ADM with the balanced Add/Drop action ($L_d^{in}$ = 6.9 mm) under the In port launching (Plane 1) and the Add port launching (Plane 2) conditions. **c** Power transmittances of the two grating-loaded side couplers at the Through port (Plane 2) under the In port launching (Plane 1) condition. Inserts schematically show the signal optical paths corresponding to various ADM actions.

launching port with only minimal back-reflections and cross talk (smaller than −30 dB by power for any port). When integrating Y-splitters into ADM circuits, their contribution to the cross talk is negligible, with the main negative effect being additional propagation loss incurred by the signals propagating through the curved arms. The Y-splitters that were employed in this work can be replaced by other compact and efficient beam steering solutions. Therefore, instead of using the actual powers at the different ports of ADM circuits after propagation through Y-splitters, it is more convenient for the design purpose to use powers right at the output of grating-loaded side coupler element (Planes 1 and 2), which can be estimated by using Eq. (1) in numerical simulations (see Supplementary Note 7 for detailed information).

It should be noted that the relative position of a 20.6-mm-long grating inside the 35-mm-long side coupler section (controlled by the $L_d^{in}$ parameter) gives rise to different interference conditions between the two supermodes of a coupler, which we exploit to find optimal coupler designs. Particular attention is paid to the design and characterization of two grating-loaded side coupler sections that enable ADM circuits with either an optimized Drop action or a balanced Add/Drop action.

To optimize the Drop action we choose $L_d^{in}$ that results in the highest power at the Drop port (Plane 1) for the carrier frequency at the center of the grating stopband. Defining $L_g$ to be the light penetration distance into grating, the optimal position of the grating section $L_d^{in}$ can be written as:

$$L_d^{in} + L_g \approx L_c/2 \qquad (3)$$

Under this condition, the destructive interference of the two reflected supermodes at the In port (Plane 1) ensures the maximal power of the THz signal at the Drop port (Plane 1)[17,46]. Another way to optimize the ADM design is to reduce the difference between spectral performances of the Add and Drop actions at the expense of their absolute performance, thus resulting in a balanced ADM performance. That means that a similar output power will be registered at the Drop (Plane 1) and the Through (Plane 2) ports when the broadband THz signal is launched into the In (Plane 1) or the Add (Plane 2) ports correspondingly. In this optimization, one minimizes the difference between transmitted powers in the Through port under Add port launching and in the Drop port under the In port launching.

By comparing the ADM circuits featuring different $L_d^{in}$ (see Supplementary Note 7), the grating-loaded side couplers featuring $L_d^{in}$ of 1.5 and 6.9 mm are chosen to enable the ADM circuits with optimized Drop action and balanced Add/Drop action, correspondingly. Figure 4b–e shows the numerically computed

power transmittance and reflectance of these two grating-loaded side couplers when the light is launched into the In port (Plane 1) and the Add port (Plane 2). We note that $L_g$ and $L_c$ parameters are frequency-dependent across the grating stopband. Therefore, ripples in transmittance and reflectance spectra of various ports are observed even for these optimal designs.

These two ADM designs were then experimentally characterized. We present the normalized power transmittances at Planes 1 and 2 to characterize the performance of the grating-loaded side coupler section while decoupling it from the losses incurred in Y-splitters of an ADM. By using lower loss splitters one can significantly improve the loss characteristics of ADM devices, which are ultimately limited by those of the grating-loaded side coupler sections. Therefore, further research into the development of the more performant Y-splitters is in order, which, however, is beyond the scope of this paper.

In Fig. 5a, we present experimental and numerical transmittances at Drop port (Plane 1) under the In port launching (Plane 1) and transmittances at the Through port (Plane 2) under the Add port launching (Plane 2) of the grating-loaded side coupler that enables the ADM circuit with optimized Drop action. While an overall good correspondence between numerical and experimental data is observed, we also note that experimental transmittances show more pronounced ripples in their spectra, as well as somewhat lower bandwidths. Nevertheless, the relatively smooth 12-GHz-bandwidth peak-shaped power transmittance spectrum confirms the efficient channel Drop action of this ADM for THz carriers that fall within the whole grating stopband.

Discrepancies between the results of numerical analysis and experimental characterization of a grating-loaded side coupler stem from several factors. The geometrical deviation of the fabricated Y junction from the theoretical one (see Fig. 3b) results in a somewhat modified value of the coupling length $L_c$, thus leading to a discrepancy in the numerically predicted and experimentally measured performance of the side couples (see Fig. 3e). Moreover, geometrical nonuniformity of the fabricated Bragg gratings leads to higher propagation losses, smaller grating strength, and longer grating penetration depth $L_g$, thus changing the resonant condition given by Eq. (3) and leading to a discrepancy between the numerical and experimental Add/Drop spectra shown in Fig. 5. The inconsistency between numerical and experimental transmittances becomes pronounced for the grating-loaded side coupler designed for the ADM circuit with balanced Add/Drop action (see Fig. 5b). Nevertheless, it is noted that similar transmittances were measured at the Drop port (Plane 1) for channel dropping and the Through port (Plane 2) for channel adding in experiments. Furthermore, Fig. 5c shows that the measured power transmittances at the Through port (Plane 2) under

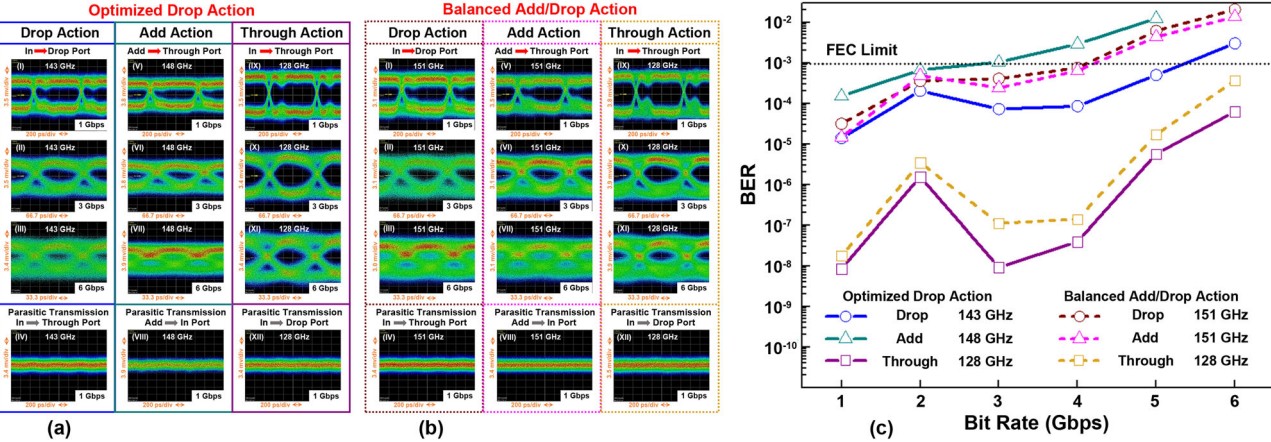

**Fig. 6 Experimental study of the ADM circuits for THz communications.** Eye patterns for the THz signals of different bit rates and different carrier frequencies propagating through **a** an ADM circuit with the optimized Drop action and **b** an ADM circuit with the balanced Add/Drop action. The measurements were performed at the following ports: (I–III) Drop port, (IV) Through port (Drop action: In port launch within grating stopband); (V–VII) Through port, (VIII) In port (Add action: Add port launch within grating stopband); (IX–XI) Through port, (XII) Drop port (Through action: In port launch within grating passband). The measurements are performed at various frequencies in the grating stopband (143, 148, 151 GHz) and the grating passband (128 GHz), which are also marked by black stars in Fig. 5. **c** Measured BER versus bit rate of the THz carrier wave propagating through two optimized ADM circuits.

the In port launching (Plane 1) condition for these two grating-loaded side couplers in optimal designs are similar.

**Characterization of the ADM circuits using a THz communication system.** Finally, we experimentally characterized the two complete THz ADM circuits, one with optimized Drop action ($L_d^{in} = 1.5$ mm) and the other one with balanced Add/Drop action ($L_d^{in} = 6.9$ mm) whose insertion loss for THz light registered at different ports are shown in Supplementary Notes 8 and 9, using an in-house photonics-based THz communication system with amplitude-shift-keying modulation format. The eye patterns were recorded at different ports of ADM circuits for THz signals within the grating stopband (Add/Drop action), as well as within the grating passband (Through action) for various data rates as shown in Fig. 6a and b. Due to the relatively high insertion loss of ADM circuits and the low power of the optical THz source, the operational data rates are limited to ~6 Gbps. Also, to confirm the key role of the WBGs for the functioning of ADM circuits, a supplementary experimental study of the directional coupler circuit without a grating is presented in Supplementary Note 10.

For the ADM with optimized Drop action, the minimal insertion loss (~14 dB) for channel dropping is observed in the middle of the grating stopband at 143 GHz. Therefore, when the THz signal with the carrier frequency of 143 GHz is launched into the In port, the eye pattern recorded at the Drop port shows the highest eye amplitude within the grating stopband (see Fig. 6a I–III), while the eye diagram for the parasitic transmission into Through port is shown in Fig. 6a IV. As only the Drop action has been optimized, the performance of the Add action of this ADM is not symmetric with respect to that of the Drop action. Thus, the highest efficiency of the Add action is achieved within the grating stopband at 148 GHz (insertion loss of ~18 dB). In experiments, the THz signal was launched into the Add port and measured at the Through port for different data rates (see Fig. 6a V–VII), while parasitic transmission into the In port is shown in Fig. 6a VIII. The corresponding BERs for the added (Through port) and the dropped (Drop port) signals are shown in Fig. 6c, from which it is clear that a nonsymmetric ADM also features nonsymmetric BER performances for the Add and Drop actions.

For the THz ADM with balanced Add/Drop performance, the lowest insertion loss is at 151 GHz within the grating stopband. Unsurprisingly, the resultant eye patterns for the Add and Drop functions are similar and so are the corresponding BER curves (see Fig. 6b I–III & V–VII and c). In Fig. 6b IV and VIII we also show the parasitic transmission in the Through port for the Drop action and in the In port for the Add action, correspondingly.

Despite the difference in the channel dropping and adding functions of these two ADMs, the Through function of both ADMs in the grating passband is remarkably similar. This is manifested by the similar eye patterns (see Fig. 6a and b IX–XI) and BER curves (see Fig. 6c) for the THz signals within the grating passband (ex. 128 GHz). Such similar performance in the passband is expected as the main difference between these two designs is the relative position of gratings within the side coupler sections of a fixed length. In the passband, though, transmitted light will only see the total length of the coupler circuit, while only being slightly modified by the presence of gratings. Parasitic transmissions into the Drop port when the signal in the grating passband is launched into the In port for Through action are shown in Fig. 6a and b XII. It is found that the cross talk for the Add/Drop function in the stopband and Through function in the passband is below the system noise level.

Finally, we present the BER response of these two ADM circuits (see Fig. 6c). In the analysis, we have to distinguish the Through action in the grating passband from the Add/Drop actions in the grating stopband. For the Through action, it is the dispersion and transmission losses of the straight waveguide sections along the In–Through path of ADMs, as well as dispersion and losses of the Bragg gratings in the passband that limit the data rates. For the Add/Drop action, it is the dispersion and losses in the straight and curved branches of a Y-splitter, as well as dispersion and losses of the Bragg gratings in the stopband that limit the data rates. Losses of various ADM subcomponents (quoted by power in the following discussion) affect significantly and unequally the BER of the Add/Drop and Through actions due to the negative effect of the noise on the detection accuracy. The highest loss in our device ~8 dB is a bending (radiation) loss of the 6.3-cm-long curved branch of a Y-splitter that mostly affects the Add/Drop action. The grating-loaded side coupler

section results in ~7 dB loss for operation in the grating passband (Through action), while resulting in over 4 dB loss for operation in the grating stopband (Add/Drop action). These losses are due to several factors including ohmic losses, the mismatch between the modal fields of a straight waveguide and a grating, as well as scattering on the imperfections in the grating structure. Next, there is butt-coupling loss of ~3 dB at each interface between ADM and the waveguide-coupled THz emitter and detector[29]. Finally, there are relatively low losses of straight waveguides (0.4 dB/cm), which are mostly due to ohmic losses incurred by the plasmonic modes in the silver layers. As a result, the relatively high insertion losses of the developed ADMs (~11 dB for the channel Through and >14 dB for the channel Add/Drop actions) and the coupling loss between ADM and THz emitter and detector (~3 dB at each interface), together with the low power of the employed THz emitter (−9 to −6 dBm in the 120–160 GHz spectral range) exceed the limit for the error-free data transmission, which for our THz communication system is ~−20 dBm[54]. This results in the relatively low signal-to-noise ratio during BER measurements (see Supplementary Note 11) and, as a consequence, high and unequal BERs for the channel Add/Drop and Through actions.

It is also important to comment on the potential effect of the group velocity dispersion on the ADM performance (see Supplementary Note 12 for details) as strong modal dispersion can lead to signal scrabbling that manifests itself in the eye diagram distortion and closing of the eye pattern. In our case, however, both straight and bent waveguides are short (~10 cm) and feature relatively small dispersions (~3 and 10 ps/(THz*cm), respectively), therefore, the impact of modal dispersion on the quality of transmitted data is relatively small compared to the effect of ADM losses discussed earlier. A detailed analysis shows that, ultimately, the Through action is indeed limited by the dispersion of a straight waveguide, with maximal bit rates in the 10 s of Gbps. At the same time, the Add/Drop action is rather limited by the size and shape of the grating stopband, which in our case limits the bit rates to ~4–8 Gbps.

## Discussion

We demonstrated four-port ADM circuits capable of channel adding and dropping for applications in THz communication systems operating within the FDM modality. We used 3D-printed two-wire plasmonic waveguide as a base technology to develop reconfigurable plasmonic circuits comprising WBGs, directional couplers, grating-loaded side couplers, and Y-splitters. We believe that the air-core two-wire waveguides offer many advantages for THz circuit development, including low losses due to the high modal presence in the air, device tunability due to ease of access to modal fields, 3D integration potential, modular nature, low coupling loss between modules, as well as the capability for robust and cost-effective prototyping and fabrication. Two ADM types were studied, one is the Drop action optimized design, the other one is the balanced Add/Drop design, both featuring experimental bandwidths of 12 GHz (grating stopband) in the vicinity of 143 GHz. Mainly limited by the low power of the THz signal received by the receiver in the photonics-based THz communication system, the Drop action with data rates of up to 5 Gbps at 143 GHz carrier and the Add action with data rates of up to 3 Gbps at 148 GHz carrier for the Drop optimized ADM circuit were confirmed to result in bit error rates lower than $10^{-3}$ (forward error correction (FEC) limit). For the balanced Add/Drop multiplexer design, both Add and Drop actions supported data rates of up to 4 Gbps at 151 GHz carrier with BER < $10^{-3}$. The Through action in the grating passband (ex. 128 GHz) was shown to support bit rates

exceeding 6 Gbps for both designs. Featuring low GVD of the underlying waveguides, the error-free data transmission of THz signal processed by these two ADMs is expected using THz transmitters with high power.

The main challenge for the proposed ADM designs was relatively high insertion losses (~11 dB for the Through action and larger than 14 dB for the Add and Drop actions) together with the butt-coupling loss of ~6 dB. The main reasons for such losses, in order of importance, are suboptimal Y-splitter (~8 dB), insertion losses of Bragg gratings, and scattering in gratings due to fabrication imperfections (~7 dB for grating passband and >4 dB for grating stopband), butt-coupling loss at each interface (~3 dB), transmission losses of straight two-wire waveguides (~0.4 dB/cm). Further optimization of the ADM circuits is possible by employing more efficient Y-splitter designs and resorting to better fabrication strategies, which include using higher-resolution 3D printers and more advanced metal deposition techniques for metallization. In addition, coupling losses to detector and emitter can be further reduced via geometry optimization of the in-coupling and out-coupling ports via mode shaping. We estimate that a combined effect of such optimizations could reduce the insertion loss by 5–10 dB, while also improving and balancing the performance of the Add and Drop actions, thus rendering ADM devices highly suitable for the ultra-fast analog signal processing in THz communications.

## Methods

**Fabrication of the two-wire waveguide components**. For the ease of fabrication, each proposed micro-encapsulated two-wire plasmonic waveguide component such as straight and curved waveguide section, WBG module, directional coupler, and grating-loaded side coupler, is split into two complementary parts. Each part comprises one or two wires (depending on the waveguide component) that are attached to a half cage using deeply subwavelength dielectric support ridges of width 150 μm (see Figs. 1, 2a, and 3a)[29]. The structures are fabricated using an SLA 3D printer (Asiga® Freeform PRO2) featuring an XY-axis resolution of 50 μm. The waveguide direction corresponds to the Z-axis of the 3D printer. To ensure high geometrical precision, the parts of the waveguide component with periodic variation in the cross-sections were printed using the finest resolution in the Z-axis (layer thickness of 10 μm). The parts with smooth surfaces were fabricated with the Z-axis resolution of 25 μm to reduce the fabrication time. After fabrication, the inner surface of the cage was covered using masking tape leaving the plastic wire support uncovered. Then, a silver layer was deposited on top of the plastic wires using wet chemistry deposition to form conductive surfaces. Finally, the two-wire waveguide components were obtained by assembling the two selectively metalized parts by sliding the corresponding parts into each other using V-grooves and V-ridges that were printed into the cage. The complete THz circuits (directional coupler and ADM) were then assembled from the individual subcomponents using another set of alignment and connectorization elements that are imprinted onto the cage end-facets for seamless connectorization (see Figs. 1 and 3b).

**Characterization of the waveguide Bragg gratings**. The two-wire waveguide components and circuits were characterized using a network analyzer (N5247B PNA-X, Keysight) together with the frequency multiplier modules (Vector Network Analyzer WR5.1 and WR8.0 Extenders, Virginia Diodes).

The WBGs were characterized as follows. First, a two-wire waveguide section featuring the same 2.5 cm length as the WBGs was placed between the two two-wire waveguide sections (see Fig. S3). This assembly was then butt-coupled on both sides to the rectangular waveguides of the two extender modules. Then, the $S_{r,21}^2$ (transmission by power) of the assembly was measured as a reference. Next, the 2.5-cm-long waveguide section was replaced by the WBG element, and the $S_{21}^2$ and $S_{11}^2$ of the assembled waveguide component were recorded. The relative transmittance of the WBG was obtained by comparing the measured $S_{21}^2$ value with that of the reference. Finally, the WBG element was removed, and a planar metallic mirror was placed in its place, while the $S_{r,11}^2$ was remeasured to serve as a reference for the WBG reflectance estimation. To remove oscillations from the reflected spectra, we used the midpoint between the adjacent maxima and minima of the recorded reference $S_{r,11}^2$ to fit the reference spectra with a smooth curve. Then, we normalized the WBG reflectance by dividing the midpoint between the adjacent maxima and minima on the recorded WBG spectra $S_{11}^2$ by the fitted reference to obtain the data shown in Fig. 2d (see Supplementary Note 4 for more details).

**Characterization of the side couplers**. The THz side coupler containing two two-wire waveguides was characterized as follows. First, the side coupler was placed between two Y-splitters (see Fig. 3b) and then inserted into the spectroscopy system. The THz signal was launched into IP 1 and the transmitted powers received at OP1 and OP 2 were recorded. The power of THz light in the Waveguide 1 and 2 at the output plane of the side coupler section (Plane 2) (see Fig. 3b) was obtained by dividing the transmitted power recorded at OP 1 and OP 2 by the insertion loss of the straight and curved two-wire waveguide in the two arms of Y-splitter (see Supplementary Note 5), correspondingly. The power ratio of each waveguide was then computed by dividing its derived output power by the sum of the two values. By using side coupler sections of different lengths and after repeating the measurements, the power ratios in the two waveguides as a function of a side coupler length were obtained.

**Characterization of the grating-loaded side couplers**. The power transmittance of the grating-loaded side couplers was characterized as follows. The grating-loaded side coupler section was placed between two Y-splitters. First, to measure the transmittance of the Drop port under the In port launching, the ADM circuit was placed between extenders with the In and Drop ports butt-coupled to the rectangular metallic waveguides. The transmitted power spectrum ($S_{21}^2$) of the Drop port was recorded when launching the THz light into the In port. Next, the ADM circuit was replaced by a two-wire waveguide assembly comprising straight and curved waveguide sections of the same lengths and geometry as straight and curved arms of a Y-splitter. Its transmitted power spectrum ($S_{21}^2$) was recorded and used as a reference. Particularly, the transmittance of the Drop port at Plane 1 of the grating-loaded side coupler was obtained by dividing the measured transmitted power spectrum at the Drop port of the ADM by the reference spectrum, thus decoupling the ADM response from the losses incurred in the Y-splitter. The transmittance of the Through port of grating-loaded side coupler under the Add port launching condition at Plane 2 for channel adding was characterized using a similar procedure after connecting the Add and Through ports with extenders.

The transmittance of the Through port at Plane 2 of the grating-loaded side coupler section under the In port launching (Plane 1) condition was measured by placing the THz ADM circuit between the two extenders with its In and Through ports butt-coupled to the metallic rectangular waveguides. Then the transmitted power spectrum ($S_{21}^2$) of the Through port was recorded under the In port launching condition. Next, the transmitted power spectrum ($S_{21}^2$) of a reference waveguide comprising two sections with lengths and geometries identical to those of the straight arms of the Y-splitters was recorded. The transmittance of the Through port at Plane 2 was obtained by dividing the transmitted power spectrum recorded at the Through port of ADM by the reference.

**Characterization of the ADM circuits using a THz communication system**. The two-wire plasmonic ADM circuits were also characterized using a photonics-based THz communication system. The schematic diagram of the experimental setup is shown in Supplementary Note 13 and briefly explained in what follows. In the transmitter section, two independently tunable DFB lasers (Toptica Photonics) operating in the infrared C-band with slightly different center frequencies are combined using a 3 dB coupler as the source of THz generation. The baseband signal source of pseudorandom bit sequence (PRBS) with a varying bit rate from 1 to 6 Gbps and pattern length of $2^{31} − 1$ is generated by the pulse pattern generator (PPG) unit integrated into the test equipment (Anritsu-MP2100B). The baseband signal is then amplified using the RF amplifier (Thorlabs-MX10A) which drives the Mach–Zehnder modulator (Thorlabs-LN81S-FC) to modulate the intensity of laser beams. Then the modulated laser beams are amplified using an erbium-doped fiber amplifier (EDFA) (Calmar laser-AMP-PM-18) and injected into a waveguide-coupled uni-traveling-carrier-photodiode (UTC-PD) photomixer (NTT Electronics) to generate modulated THz carrier wave with an operation frequency corresponding to the beat frequency of the two tunable DFB lasers. In the receiver section, the THz carrier wave is detected and demodulated using a zero bias Schottky diode (Virginia Diodes-WR6.5-ZBD-F), and then amplified using a high gain low noise amplifier (LNA) (Fairview Microwave-SLNA-030-32-30-SMA). Finally, the eye pattern and BER are recorded using the test equipment (Anritsu-MP2100B)[54,55].

In experiments, the THz carrier wave within the spectral range of 110–170 GHz having the power in the range between 125 μW (−9 dBm) and 250 μW (−6 dBm) was butt-coupled into the THz ADM circuit via a 1-inch-long WR6.5 rectangular waveguide (Virginia Diodes WR6.5). A similar arrangement was used to connect the output port of the THz ADM circuit to the Schottky diode (see Supplementary Note 13). Three operation modalities of an ADM circuit (Add, Drop, Through) were studied with the transmitter and receiver connected to Add and Through ports, In and Drop ports, and In and Through ports, respectively. In each case, we recorded the eye patterns for the THz signals of different bit rates and carried out the corresponding BER measurements. It is noted that, during BER measurements, the decision threshold was optimized to equalize the insertion error (digital 0 is mistaken for a digital 1) and the omission error (digital 1 is mistaken for a digital 0). It is then recorded within the duration of 1/(target BER × bit rate), where the target BER was set to $10^{−12}$ (error-free).

**Numerical simulations**. The numerical studies of the proposed WBGs, directional coupler circuits, and ADM circuits were carried out using the commercial finite element software COMSOL Multiphysics within the 3D finite element frequency domain module with ports. Within this formulation, transmittance and reflectance can be characterized by using scattering matrix coefficients related to each port. In our simulations we used frequency-dependent refractive index ($n_{resin}(f) = 1.654−0.07f$ [THz]) and material absorption ($\alpha(f)$ [cm$^{−1}$] = 0.64 + 13.44$(f$ [THz]$)^2$) of resin (used in 3D printing) in THz spectral range of 0.1–0.3 THz, which were obtained in a prior experimental study detailed in ref. [56]. The metallic wires were modeled using impedance boundary condition (IBC) at the wire surface together with the Drude-Lorentz model for the dielectric constant of silver obtained from refs. [29,57].

$$\varepsilon_m = 1 - \frac{\omega_p^2}{\omega^2 + i\omega\gamma_b} \tag{4}$$

where $\omega_p = 2\pi \cdot 2.185e15$ Hz is the plasma frequency of silver, and $\gamma_b = 2\pi \cdot 2.69e14$ Hz is the fitted value of the damping coefficient of the Ag layer deposited on resin support in THz spectral range.

Due to the geometric symmetry of the two-wire WBG module, only half of this structure together with perfect electrical conductor (PEC) boundary condition was used in numerical simulations. When the THz light is launched using port boundary condition at the input facet of the WBG, the reflectance at the same port and transmittance at the port on the other end facet were computed (see Supplementary Note 14 for example) using the $S_{11}^2$ and $S_{21}^2$ coefficients.

By following a similar procedure discussed above, the directional coupler and ADM circuits were studied using numerical simulations. Different from the WBG models, the full asymmetric 3D structures of the directional coupler and ADM circuits were used. The scattering matrix coefficient at the four defined ports is computed under various launching conditions, i.e., the OP 1 and OP 2 under the IP 1 launching condition for directional coupler circuit (see Fig. 3d), as well as the In, Drop, Add, and Through ports under the In port launching and the Add port launching conditions for ADM circuit (see Fig. 4a). The transmittance and reflectance (by power) of side couplers at Planes 1 and 2 were then obtained using Eq. (1).

## Data availability
The authors declare that the main data supporting the findings of this study are available within the article and its Supplementary Information files. Extra data are available from the corresponding author upon request.

## Code availability
The authors declare that the description of numerical simulations has been provided in the article and its Supplementary Information files. Any simulation and computational codes for this study are available from the corresponding authors on reasonable request.

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

## Acknowledgements

This work was supported by Canada Research Chair I in Ubiquitous THz photonics (34633) awarded to M.S.; Y.C. acknowledges support from China Scholarship Council (201706250016). The authors would like to thank the Centre for Advanced Research in Microwave and Space Electronics (POLY-GRAMES) of Polytechnique Montréal for the use of the Network Analyzer.

## Author contributions

Y.C., K.N., G.X., and M.S. conceived these experiments and contributed to their design. Y.C. and K.N. performed the numerical simulations and measurements. Y.C., K.N., and M.S. contributed to writing the manuscript.

## Competing interests

The authors declare no competing interests.
