## [Peer Review File · Nature Communications]

REVIEWER COMMENTS

Reviewer #1 (Remarks to the Author):

The authors present a mm-wave add-drop multiplexer that is implemented in a two-wire waveguide structure. A pair of such waveguides is brought together using a Y-junction in order to form a directional coupler, and in the centre section, the width of the individual conductor lines is modulated with a periodically ribbed structure that serves as a Bragg mirror. This yields frequency selectivity; the add/drop band is reflected during coupling, whilst other frequencies pass through. The intricate ribbing structure is made possible with 3D-printing followed by metallization. It is easy to imagine large-scale networks of add-drop multiplexers of this sort, where each element contains a Bragg mirror that reflects a different stopband.

My recommendation is a major review. Please address the following itemized issues.

1. In Figs. 2 and 5, the agreement between measurement and expectation is not especially close, and hence the intended functionality is not adequately validated. For electromagnetic device design, higher-quality experimental results are needed to merit publication as a journal article.
2. The core concept appears to be related to [R1]; a Bragg mirror is embedded into a four-port directional coupler, and the stopband is rejected, forming the add/drop bandwidth. The referenced device is implemented at nanoscale, for telecom wavelength, in contrast to the two-wire-waveguide mm-wave device under review. But aside from this, is there any additional novelty? The fact that the present device is 3D-printed (i.e. as opposed to nanofab) is not sufficient, as fabrication quality is given as a reason for deviation from expected performance on lines 144-146: "... experimental WBGs show somewhat shallower transmission spectra compared [sic] to the numerical simulations, which is attributed to imperfections in the geometrical structure of 3D printed gratings." The novelty of a given fabrication methodology is closely linked to the quality of the final result.
3. In the communications experiment that is presented in Fig. 6, a clear channel is not obtained for the add and drop signal paths, which exhibit BER>10⁻⁵ even for a relatively low data rate of 1 Gbit/s. This is a thousand times more bit errors than for the through channel. Is there some additional dispersion or reflection in the add/drop paths?
4. The authors have employed Y-junctions to interface the individual ports with the Bragg mirror. However, an ideal Y-junction exhibits 3 dB of loss when splitting, and also when joining, due to reciprocity. All signal paths through this device (i.e. add, drop, and through) must pass through two Y-

junctions, for a minimum of 6 dB loss, or a maximum efficiency of 25%. Can the authors justify this innately low efficiency?

5. What is the size of the overall device? And how does it compare to contemporary MMIC systems?

6. Is this device compatible with standard mm-wave connectors? If so, why have the authors chosen to employ a lens-coupled photomixer that is intended to interface with free space? This choice must be justified, as the authors directly attribute spurious features of the measured results to this photomixer in lines 157-159: "...strong ripples that are presented in the experimental reflection spectra (see Fig. 2(d)). These result from the standing waves formed in the cavity of the CW-THz spectroscopy setup, as well as inside of a photomixer silicon lens."?

7. This is a minor issue, but the quality of English language could be improved. For instance, there are errors in the abstract: "Terahertz band is considered as..." -> "The terahertz band is considered to be...", and "...attention is payed..." -> "...attention is paid...". I recommend having a native speaker of the English language review the manuscript.

[R1] G. Biagi et. al. "Compact wavelength add-drop multiplexers using Bragg gratings in coupled dielectric-loaded plasmonic waveguides," Opt. Lett. 40, 2429-2432 (2015)

Reviewer #2 (Remarks to the Author):

This paper shows fabrication and characterization of a 3d printed add-drop multiplexer for THz communications using fiber Bragg grating. This subject is of relevance since the THz band is the future of wireless communications. To reach the goal of truly functioning THz systems, signal processing devices are required. Here, the researchers demonstrate a 3d printed add-drop multiplexer. This work is novel in the strategy used and can definitely attract the interest of the scientific community working in wireless communications at high frequencies. The design strategy is well documented with both simulations and experiments, which are carried using both a THz spectrometer and a communications system. The conclusions are well supported, and all the relevant details are provided in the supplementary (which is something that is not usually seen with such detail). I have however multiple comments about the general presentation. Reading the manuscript is tedious and some of the figures are hard to understand (see detailed comments below). Based on these comments, I believe this paper can be accepted if the researchers correct the following points, which I consider minor because they are mainly related to the format:

- 1) Abstract. line 13: payed -> paid. Line 16: that THE proposed
- 2) In the introduction, optical networks are compared to terahertz networks. However, the mentioned optical networks operate in fiber, whereas the comparison should also be done with microwave/RF WIRELESS communications.
- 3) Line 70: drop/add -> add/drop
- 4) Line 71-74: It is mentioned that thermal and mechanical tuning of the proposed circuits is possible. First, how? Second, thermal and mechanical tuning are not preferred for dynamic band allocation. How are the proposed circuits better than the ones based on a silicon platform, since these can be modulated with charge injection? How something similar can be done with a purely metallic waveguide?
- 5) Fig. 2a: Add H and Lambda in the figure when presenting the geometry.
- 6) Fig. 2b: It is hard to decipher the figures as there are too many curves, and there are not distinguishable (the squares and triangles look the same). I suggest the researchers consider splitting the individual figures in 2 (one on top and one on bottom). It is absolutely hard to understand in its current form. The same comment can be made for Fig. 2cd.
- 7) Line 161: ...notoriously difficult to eliminate. Needs a reference.
- 8) Fig. 3a and 4a. The figure is confusing, the through beam should go from left to right.
- 9) Lines 280-290: The difference between Balanced and Optimal designs is not clear, should be emphasized more, especially since it becomes important later in the text.
- 10) Line 301: While in this work, we use relatively lossy Y splitters comprising straight and curved branches. This is not a sentence.
- 11) Fig. 5: It is hard to follow where the inputs and outputs are. Small schematics showing In and Out ports for the figures should be added in all subpanels.
- 12) Line 433: It is mentioned that better fabrication strategies can be used. Would making everything in metal rather than 3d printing be better?
- 13) The legends of all the figures are too long and hard to follow. They should be kept simple and straight to the point with minimal detail (especially since it is sometimes the same geometric details in all the subpanels).
- 14) I am wondering if using two bare waveguides (with no gratings) would show similar performance in communications? In short, how are the authors convinced that what they are observing experimentally is really due to the effect of their gratings.

Reviewer 1:

First, we would like to thank the reviewer for their time and thoughtful comments. To address the reviewer's concerns, we redid the experimental characterization of our ADM components using a solid-state THz spectroscopy system that features a more powerful continuous-wave THz source compared to the original experiment where we have used an optical-based continuous-wave THz spectroscopy system. We have also modified the figures and text to address the reviewer's concerns.

For convenience, we indicate in **bold** the original questions by the reviewer, in regular font our response to the reviewer, and in *italic* our corrections to the manuscript.

The authors present a mm-wave add-drop multiplexer that is implemented in a two-wire waveguide structure. A pair of such waveguides is brought together using a Y-junction in order to form a directional coupler, and in the centre section, the width of the individual conductor lines is modulated with a periodically ribbed structure that serves as a Bragg mirror. This yields frequency selectivity; the add/drop band is reflected during coupling, whilst other frequencies pass through. The intricate ribbing structure is made possible with 3D-printing followed by metallization. It is easy to imagine large-scale networks of add-drop multiplexers of this sort, where each element contains a Bragg mirror that reflects a different stopband.

My recommendation is a major review. Please address the following itemized issues.

The following are our replies to the reviewer queries:

Query 1. In Figs. 2 and 5, the agreement between measurement and expectation is not especially close, and hence the intended functionality is not adequately validated. For electromagnetic device design, higher-quality experimental results are needed to merit publication as a journal article.

Q1. Author Response:

We agree with the reviewer that the experimental results for the fabricated two-wire waveguide components differ somewhat from the numerical simulations, however, this statement has to be nuanced. Moreover, we argue that such a discrepancy is unavoidable due to standing wave effects when performing CW THz spectroscopy (regardless of the source power), which is difficult/impossible to mitigate in the absence of THz isolators.

In the revised work, to increase the quality of optical characterization and reduce the effect of noise on the system performance, we redid the optical characterization of all the THz circuits using a network analyzer and a solid-state THz source. Compared to the optics-based CW-THz spectroscopy system used in the original publication, the network analyzer enables measurement with a much higher SNR due to higher output powers of the generated THz wave (~0 dBm within the studied spectral range), fine spectral resolution (~0.02 GHz), as well as no standing waves inside the solid-state THz emitter and detector modules in contrast to the photomixer modules (see more details in our answer to your Query #6). The newly obtained experimental results for the two-wire WBGs and grating-loaded side couplers show very similar spectral responses to the ones obtained using an optics-based system, however with dramatically reduced noise contribution, which can be seen in the revised Figs. 2 and 5. For the sake of completeness, in the main paper, we now present the results of spectral characterization using a network analyzer, while we still present the original results obtained using an optics-based CW spectroscopy system in Supplementary Note 9. At the same time, the BER measurements were still performed using an optics-based system as we could not find any solid-state system in the Montreal area that could perform such measurement at ~150 GHz.

In what follows we concentrate on the reasons for the discrepancy between the experimental and theoretical data, and question whether such a discrepancy is indeed “pronounced” as stated by the reviewer.

Firstly, we note that most key features of the WBG experimental performance show good correspondence with the numerical simulations (see Fig. 2). These include the center frequency of the WBG stopband, the size of a stopband, as well as the minimum/maximum values of the transmission/reflection coefficients for the WBG stopband. In fact, the main difference between

experiment and theory is in the WBG reflection spectra. When characterizing WBG reflection spectra using the optics-based CW-THz spectroscopy system, this difference is mainly due to a combination of the low signal to noise ratio (SNR) of the indirect measurement assisted by a lossy Y-coupler, and the effect of standing waves in the photomixer lens and inside of the coupler-WBG arrangement (see Fig. S13(c)). In the absence of THz isolators, selective attenuation of the parasitic back-reflected waves is challenging (if not impossible), which sets a fundamental limitation on the quality of comparison between the experimental and theoretical results.

When characterizing the WBG reflection spectrum using the network analyzer, the reflection coefficient of a WBG can be measured directly using an S_{11} parameter without resorting to the indirect Y-coupler-assisted measurement. Unfortunately, in this measurement one still observes significant contribution of the standing waves, which are caused by the parasitic reflections between various components and detector, thus, significantly complicating interpretation of the measured results due to periodic ripples in the measured spectra. Nevertheless, by comparing the measured S_{11} spectra with the theoretical predictions of reflection (see Fig. 2(d)), one concludes that the fabricated WBGs indeed feature high reflection within a stopband, with both the stopband size and the absolute values of reflection matching well the theoretical predictions.

Secondly, when integrating the WBGs into ADMs and characterizing the grating-loaded side couplers using the network analyzer (Fig. 5), we note that the center operation frequencies of the Add and Drop channels and spectral response of the Through channel correspond well to the numerical simulations. At the same time, the bandwidths and the spectral shapes for the Add/Drop channels indeed differ somewhat from the theoretical predictions. Such disagreements are also present in the corresponding spectra measured by the CW-THz spectroscopy system (see Fig. S14). This discrepancy, however, does not negate the functionality or utility of the proposed devices, as it is simply indicative of challenges in the proper characterization of such components (the lack of THz isolators, etc.), and the necessity of further refinement of the numerical model to reproduce all the finer details of the ADM spectral response. As a result, a reasonable but imperfect correspondence between theory and experiment is still indicative of a strong potential of 3D printing approach for the development of components for THz communications, while it also clearly indicates that a significant refinement of the measurement and fabrication methods are still needed for this methodology to find its place in the commercial applications. We, therefore, believe that one of the key contributions of this work is in setting up a benchmark for 3D printing techniques in application to complex component fabrication for THz communications.

Q1. Author Action:

A1. We replotted Figs. 2 and 5 using the results of new experimental measurements from the network analyzer. In those figures, experimental and numerical transmission and reflection spectra of the WBGs and grating-loaded side couplers are compared, and a good-to-reasonable correspondence between the two sets is observed.

Fig. 2 Two-wire Waveguide Bragg Grating. (a) Schematic of the WBG cross-section, its side view, and a photograph of the fabricated WBG with the top half removed. The grating featuring a periodic sequence of truncated cones (light gray) and a cylindrical wire (dark gray) is suspended in the air on dielectric supports and encapsulated within a plastic cage. The numerical and experimental relative transmission coefficients (by field) of the 2.5cm-long WBG sections (b) comprising a different number of periods ($A=1.03$ mm, $H=0.21$ mm), and (c) containing 20 periods ($H=0.21$ mm) as a function of the period length A . The THz

light travels in the forward direction (see Fig. 2(a)). (d) The numerical and experimental relative reflection coefficients (by field) of the 2.5cm-long WBG sections comprising 20 periods ($A=1.03$ mm, $H=0.21$ mm) with two possible traveling directions of the THz light (see Supplementary Note 4 for more details).

Fig. 5 Optimized grating-loaded side couplers. Power transmittances of the grating-loaded side couplers for (a) an ADM with the optimized Drop action ($L_d^{in}=1.5$ mm) and (b) an ADM with the balanced Add/Drop action ($L_d^{in}=6.9$ mm) under the In port launching (Plane 1) and the Add port launching (Plane 2) conditions. (c) Power transmittances of the two grating-loaded side couplers at the Through port (Plane 2) under the In port launching (Plane 1) condition. Inserts schematically show the signal optical paths corresponding to various ADM actions.

Various parts of the paper and the Supplementary Note were modified, where appropriate, to explain the nature of the difference between the numerical and experimental results:

B1. Discussion relevant to the WBGs is placed into the third paragraph of Chapter 2.1

“We then compare the numerical and experimental results of the relative transmission coefficients for the two-wire WBGs. We first observe that a larger number of periods in a WBG results in more pronounced transmission dips (see Fig. 2(b)). We also find that the experimental WBGs show somewhat shallower transmission spectra within the grating stopbands compared to the numerical simulations. This is mainly attributed to the field attenuation and scattering in the WBG structure. Nevertheless, with only 20 periods in the WBG structure a pronounced transmission dip near 140 GHz featuring a minimal transmittance (by power) of less than 1% is found experimentally within the grating stopbands of ~ 14 GHz (FWHM) bandwidth. Furthermore, we confirm the designability of the WBG spectral response by varying grating periodicity (see Fig. 2(c)). Reducing the grating period from 1.08 mm to 1 mm shifts the center frequency of the WBG stopband from 134 GHz to 145 GHz, while measured WBG bandwidth remains largely unaffected. Additionally, both the WBG transmission and reflection spectra are virtually symmetric when switching the traveling direction of the THz light despite the WBG nonsymmetric structure (see Fig. 2(d)). The WBG's high reflection efficiency within the grating stopband was confirmed experimentally (see Supplementary Note 4), thus making them suitable for channel adding/dropping in the ADM circuits. Overall, we observe a good correspondence between the numerical and experimental spectral performance of the WBGs in terms of the center frequency, the bandwidth of the grating stopband, as well as minimal and maximal values of the normalized transmission and reflection coefficients.”

C1. We have removed from the main text the discussion of the WBG reflection coefficient measurement using an optics-based CW-THz spectroscopy system and placed it into Supplementary Note 9.

“The relative reflection coefficient of WBG (see Fig. S13(c)) was characterized by CW-THz spectroscopy system as follows. Firstly, the two-wire Y-coupler and the WBG were assembled using the interconnects at the end facets of both components. Then, two WR6.5 conical horn antennas were connected to the unused ports, namely input and output ports of the Y-coupler. The THz light was launched into the input port of the Y-coupler, guided through a curved arm of the coupler, and reflected from the WBG under study. The reflected light was then divided equally by the Y junction and directed towards both the input and output ports. The transmission spectrum of THz light at the output port was recorded. Next, the WBG was replaced by a planar metallic mirror to completely reflect the THz signal at the Y-junction (see Figs. S6(a) and S12(b)). The transmission spectrum recorded at the output port was used as the reference. Finally, the relative reflection coefficient (by field) of WBG was obtained by comparing these two measured transmission spectra. However, due to the standing waves formed in the cavity of the CW-THz spectroscopy setup and photomixer silicon lenses, the high insertion loss of the desired THz signal propagating through the assembled circuit, as well as the noise induced by the coupling between the two curved branches of Y-coupler, this measurement features low SNR, thus resulting in the experimental results differing from the numerical ones.”

D1. Discussion about experimental details and challenges related to the characterization of the WBG reflection coefficient using network analyzer is placed at the end of Supplementary Note 4, including the following figure and text:

Fig. S5 Two-wire waveguide Bragg gratings. (a) Relative transmission coefficient (by field) and (b) Measured scattering parameter (S_{11}) of a 2.5cm-long WBG section comprising 20 periods ($\Lambda=1.03$ mm, $H=0.21$ mm) for THz light with two possible traveling directions. (c) Relative reflection coefficients (by field) and (d) Measured scattering parameter (S_{11}) of 2.5cm-long WBGs containing 20 periods ($H=0.21$ mm) as a function of the period length Λ . The THz light travels in the forward direction along WBG (see Fig. 2(a)).

“...Although the proposed WBG exhibits nonsymmetric structure (see Fig. 2(a)), it, nevertheless, features similar relative transmission coefficients for the THz light traveling along either of the two directions along WBG (see Fig. S5(a)). Particularly, Fig. S5(b) shows measured reflection coefficients S_{11} for a WBG connected to the extender module via a two-wire waveguide section. The standing waves caused by the parasitic reflections between the WBG and the detector greatly complicate the interpretation of the measured results due to periodic ripples in the measured spectra. In the absence of THz isolators, selective attenuation of the parasitic back-reflected waves is challenging (if not impossible). While the reference reflection spectrum can be recorded by placing a planar metallic mirror at the output plane of the two-wire waveguide section (with WBG removed), it is still not trivial to remove oscillations from the WBG reflected spectrum. This is due to the difference in the effective light travel distance in the WBG/waveguide and mirror/waveguide (reference) measurements. Nevertheless, when comparing the normalized reflection values of the WBG to those of numerical simulations, a good comparison is found (see Fig. 2(d)). In addition, by varying the grating periodicity of WBGs, spectral positions of the reflection peaks, as well as bandwidths of the reflection spectra, agree well with the theoretical predictions (see Figs. S5(c) and S5(d)).”

E1. Discussion relevant to the ADM circuit is placed into the third paragraph of Chapter 2.3

“Discrepancies between the results of numerical analysis and experimental characterization of a grating-loaded side coupler stem from several factors. The geometrical deviation of the fabricated Y junction from the theoretical one (see Fig. 3(b)) results in a somewhat modified value of the coupling length L_c , thus leading to a discrepancy in the numerically predicted and experimentally measured performance of the side coupler (see Fig. 3(e)). Moreover, geometrical nonuniformity of the fabricated Bragg gratings leads to higher propagation losses, smaller grating strength, and longer grating penetration depth L_g , thus changing the resonant condition given by Eq. (3) and leading to a discrepancy between the numerical and experimental Add/Drop spectra shown in Fig. 5.”

Query 2. The core concept appears to be related to [R1]; a Bragg mirror is embedded into a four-port directional coupler, and the stopband is rejected, forming the add/drop bandwidth. The referenced device is implemented at nanoscale, for telecom wavelength, in contrast to the two-wire-waveguide mm-wave device under review. But aside from this, is there any

additional novelty? The fact that the present device is 3D-printed (i.e. as opposed to nanofab) is not sufficient, as fabrication quality is given as a reason for deviation from expected performance on lines 144-146: "... experimental WBGs show somewhat shallower transmission spectra compared [sic] to the numerical simulations, which is attributed to imperfections in the geometrical structure of 3D printed gratings." The novelty of a given fabrication methodology is closely linked to the quality of the final result.

Q2. Author Response:

Indeed, as mentioned by the reviewer, the working mechanism of the proposed two-wire THz ADM is not original (we never claimed that it is, actually). It was originally developed at telecom wavelengths [1] and is now cited as a reference [45] in the revised manuscript. Moreover, we note that most works that deal with communications at THz frequencies adapt the component and system-design concepts developed for either near-infrared or microwave. Therefore, as rightfully mentioned by the reviewer we need to explain the novelty and uniqueness of our contributions.

The ultimate goal of our work is to develop a cost-effective fabrication platform for high-performance, reconfigurable/tunable, and highly integrated circuits for signal processing at THz frequencies. Currently, due to relatively high losses of materials in the THz band, most of the common materials and fabrication methods are not directly applicable to the manufacturing of such THz devices. Probably the only exception is silicon photonics that can, in principle, produce low-loss THz devices, while at the same time suffering from high dispersion, the inability of 3D integration, and high fabrication costs.

In more concrete terms, the novelty and uniqueness of our approach, which is developed specifically and uniquely for the THz wave band, lies in the combination of novel plasmonic materials and structures, device fabrication strategy, subcomponent integration strategy, and cost-effectiveness.

Particularly, on the material's side, we propose using relatively cheap photosensitive resins as a support material, which is then covered with metallic layers using convenient wet chemistry techniques to enable high-performance plasmonic waveguides. Due to a large variety of such materials and processing methods, one expects that further research can greatly improve the performance and stability of such devices.

On the fabrication side, we propose using a cost-effective and fab-less 3D printing technique, which makes integrated THz devices accessible to essentially any research group, thus democratizing the field of device fabrication for the upcoming THz communications. Moreover, unique to the additive fabrication approach, 3D integrated freeform circuits are as easy to fabricate as the 2D ones, thus bringing an additional advantage of high-density out-of-plane photonic component design, which has never been explored before. We are convinced that further research in this direction could result in the first-ever demonstration of functional, truly 3D integrated terahertz circuits, that has never been done before.

Moreover, the two-wire plasmonic waveguides with air core that are in the heart of our technology are among the lowest-loss, lowest-dispersion waveguides that were ever designed for the terahertz spectral range. To date, such waveguides were only demonstrated in the form of two straight wires separated by some kind of a spacer to keep the inter-wire separation fixed, thus making such waveguides impractical for real-world applications. In this work, we have demonstrated how to adapt the concept of a two-wire waveguide to build complex and useful terahertz circuits, which is a major development in guided terahertz optics.

Additionally, air-core waveguides allow a clear path to realizing highly tunable photonic devices as the air gap size can be tuned mechanically, while the inter-wire space can be filled with liquids with tunable optical properties.

Finally, on the component integration level, the methods developed in this work show a pathway to the highly reconfigurable, modular construction of the complex terahertz circuits, which is a unique approach to integrated photonics that was not explored before.

Q2. Author Action:

A2. In the ninth paragraph in the Introduction Chapter of the main text, we made clear that the ADM design was inspired by the prior work in the near-infrared:

"We developed an interferometric THz ADM where Bragg grating is inscribed on both side-coupled two-wire waveguides, similarly to the prior work in the mid-infrared range [45-48]."

45. G. Biagi, J. Fiutowski, I. P. Radko, H. G. Rubahn, K. Pedersen, and S. I. Bozhevolnyi, "Compact wavelength add-drop multiplexers using Bragg gratings in coupled dielectric-loaded plasmonic waveguides," *Opt.*

Lett. **40**(10), 2429-2432 (2015).

46. C. Riziotis, and M. N. Zervas, "Novel full-cycle-coupler-based optical add-drop multiplexer and performance characteristics at 40-Gb/s WDM networks," *J. Light. Technol.* **21**(8), 1828 (2003).

B2. The novelty and contribution of this work are summarized in the Introduction Chapter along the lines presented in the Author Response to this Query:

"In what follows, we report design, 3D fabrication, and characterization of the four-port THz ADMs for FDM communications. The uniqueness of our approach, which is developed specifically for the THz wave band, lies in the combination of the novel plasmonic structures, device fabrication strategy, subcomponent integration strategy, and cost-effectiveness. From the point of view of fabrication, the fab-less additive manufacturing technique makes integrated THz devices accessible to anyone with a 3D printer, thus democratizing the field of device fabrication for the upcoming THz communications. Moreover, additive manufacturing allows native fabrication of the 3D integrated freeform circuits, thus opening the prospects for the high-density out-of-plane photonic integration, which is difficult to achieve with other techniques (see Supplementary Note 1 for a more detailed discussion).

More specifically, in the core of our system are micro-encapsulated two-wire plasmonic waveguides fabricated using stereolithography (SLA) 3D printing and wet chemistry metal deposition techniques [29]. Particularly, we use relatively low-cost widely available SLA printers to realize the two-wire photonic circuits featuring metalized surfaces on the resin support (see Supplementary Note 2 for this waveguide design). This fabrication route makes such circuits highly suitable for rapid prototyping and mass production. Resultant two-wire waveguides feature some of the lowest transmission and bending losses, very low group velocity dispersion (GVD), and broadband operation compared to other THz waveguides [31,32]. Such waveguides can be also easily and with high coupling efficiency butt-coupled to rectangular waveguides that are prevalent in the microwave and THz communications [29].

On the component integration level, the methods developed in this work show a pathway to the highly reconfigurable, modular construction of the complex terahertz circuits, where individual components (such as couplers, Bragg gratings, splitters, etc.) can be easily added, replaced, or subtracted to modify the device functionality.

Furthermore, thermal and mechanical tuning of the proposed circuits is readily achievable by changing the physical properties or geometry of the resin substrate and metallic film (see Supplementary Note 3 for an example). As the air-core two-wire waveguides feature a highly flexible structure and easily accessible modal fields, various elements can be inserted into the air gap to modify or tune the waveguide propagation properties. Moreover, optical and electrical tuning of such waveguides is also possible as was recently demonstrated via insertion of a graphene sheet between the two wires, which was then tuned via optical pumping or electrical gating [33-36]. Such structures promise solutions for the important problem of dynamic band allocation in the THz communication networks [37]."

29. Y. Cao, K. Nallappan, H. Guerboukha, G. Xu, and M. Skorobogatiy, "Additive manufacturing of highly reconfigurable plasmonic circuits for terahertz communications," Optica **7**(9), 1112-1125 (2020).

31. M. Mbonye, R. Mendis, and D. M. Mittleman, "A terahertz two-wire waveguide with low bending loss," Appl. Phys. Lett. **95**(23), 233506 (2009).

32. K. Wang, and D. M. Mittleman, "Metal wires for terahertz wave guiding," Nature **432**(7015), 376-379 (2004).

33. M. Chen, P. Sheng, W. Sun, and J. Cai, "A symmetric terahertz graphene-based hybrid plasmonic waveguide," Opt. Commun. **376**, 41-46 (2016).

34. W. Xu, Z. H. Zhu, K. Liu, J. F. Zhang, X. D. Yuan, Q. S. Lu, and S. Q. Qin, "Toward integrated electrically controllable directional coupling based on dielectric loaded graphene plasmonic waveguide," Opt. Lett. **40**(7), 1603-1606 (2015).

35. Z. Wang, J. Qiao, S. Zhao, S. Wang, C. He, X. Tao, and S. Wang, "Recent progress in terahertz modulation using photonic structures based on two-dimensional materials," InfoMat **3**(10), 1110-1133 (2021).

36. M. Liu, X. Yin, E. Ulin-Avila, B. Geng, T. Zentgraf, L. Ju, F. Wang, and X. Zhang, "A graphene-based broadband optical modulator," Nature **474**(7349), 64-67 (2011).

37. A. Sabharwal, A. Khoshnevis, and E. Knightly, "Opportunistic spectral usage: Bounds and a multi-band CSMA/CA protocol," IEEE/ACM Trans. Netw. **15**(3), 533-545 (2007).

C2. A detailed discussion of the novelty of this work has been added as a Supplementary Note 1 entitled "Further discussion of the novelty and key contributions of this work" accompanied with citations, and it reads as follows:

"1. Components based on the two-wire plasmonic waveguides.

High material absorption of most common materials in the THz band, as well as high waveguide dispersion, poses unique challenges for the development of THz waveguides capable of signal transmission with low insertion loss and distortion. Compared to the more common hollow metallic THz waveguides and dielectric THz fibers, the air-core two-wire plasmonic waveguides feature outstanding performance (low-loss, low-dispersion), ease of access to the modal fields, and ease of 2D and 3D on-chip integration, thus providing a promising platform for building THz circuits. The THz surface plasmon polariton (SPP)

wave can be efficiently excited by the conventional linearly polarized THz source and guided in the air between the wires with negligible loss and dispersion. Originally, the two-wire waveguides were realized using the actual metallic wires. However, due to the engineering difficulty of precise aligning and keeping the constant separation between the two wires, the resultant devices were bulky and complex. As a result, mostly the simplest components capable of basic guiding THz light over a relatively short distance were realized [1]. Somewhat more complex devices based on the two-wire waveguides were demonstrated by inserting free-standing elements in the inter-wire air gap (ex. two-wire waveguide-based antennas [2] and waveguide Bragg gratings (WBGs) [3]), while still featuring limited structural stability and reliability in practical applications. In this work, we show that by resorting to additive manufacturing one can fully exploit advanced optical characteristics of the two-wire waveguides. More precisely, we demonstrate a scalable strategy for the manufacturing of two-wire waveguide-based THz circuits based on standard optical elements such as WBGs, directional couplers, and Y-splitters. We also detail how such optical elements can be realized using two-wire waveguides augmented with some freeform elements. To our knowledge, this is the first time that the integrated optical circuits based on the two-wire waveguides are reported.

2. Fabrication route for the integrated terahertz circuits.

In our work, we pursued a combination of 3D printing and metallization techniques as an alternative to CNC machining and microfabrication for the fabrication of THz plasmonic components. In addition to being cost-efficient and requiring relatively cheap infrastructure, this fabrication route allows innovative 3D waveguide and packaging designs, as well as integration of the freeform elements, thus capable of realizing plasmonic components that are far more advanced than classic two-wire or parallel-plate waveguides [4]. Furthermore, terahertz components demonstrated in this work feature inherently modular design with the waveguides, packaging, and alignment elements all integrated into the same component in a single fabrication step. This modular architecture allows trivial assembly and reconfiguration of the complex terahertz signal processing systems without the need for painstaking alignment of the individual components. Moreover, ease of access to the modal fields in the inter-wire air gap opens new ways of dynamic tuning of the optical component performance via various means. These traits are not shared by the traditional MMICs or other reported THz waveguide components, which causes challenges when trying to integrate such stand-alone devices into larger systems.

Finally, our fabrication strategy is largely agnostic to the geometrical complexity of the optical elements, which opens a route for integration of various freeform structures and the possibility of a truly 3D component integration. For example, strong Bragg gratings integrated with two-wire waveguides are demonstrated for the first time in this work and are made possible by augmenting a two-wire waveguide with a freeform periodic cascade of truncated cones. It is important to mention that the key limitation on the quality of optical components produced by the SLA printing technique is the printer resolution. As most budget printers feature a resolution of ~ 50 microns, terahertz circuits produced by such systems are limited for operation in the lower part of the THz spectrum. However, current advances in the SLA printing hardware and photosensitive resins already demonstrated commercial systems with resolution under 10 microns, with the only limitation being the high price of such systems. We, therefore, believe that the SLA printing technique will soon be able to create integrated photonic circuits for applications anywhere in the whole THz frequency range.

3. Two-wire add-drop multiplexers (ADMs).

To date, free-space ultra-high bit rate data transmission (> 100 Gbps) using a single-channel THz link has been reported [5-7]. Enabled by THz ADMs, the THz frequency division multiplexing is expected to multiply the data-throughput capacity of the THz communication networks. However, the development of the THz ADMs is still in the early stage with only a few examples known to date, such as those based on parallel metal plates or silicon waveguides and adapting component-design concepts that are commonly used in electronics or optics (ex. leaky-wave antenna, directional coupling) [8-12]. Most reported optical-based THz ADMs are built on planar silicon substrate operating at THz frequencies ~ 300 GHz and having bandwidths of ~ 10 GHz. While functional, integration of the abovementioned ADMs into a complete THz system seems challenging as they require precise optical alignment, while some use bulky free-space components in their structure.

In our work, we report the four-port THz ADM circuits capable of channel dropping and adding the individual few-GHz-wide channels at ~ 150 GHz carrier frequency. In contrast to the prior works, we proposed a scalable and universal solution for the design and fabrication of the integrated signal processing THz circuits with an additional advantage of the modular and reconfigurable design.

In summary, tandem ADMs that allow multiple THz signals at different carrier frequencies to be selectively dropped, added, or guided through are the key enabling element of the upcoming terahertz communications networks. We believe that our integrated

circuits based on 3D printed two-wire waveguides could offer a pathway for the scalable manufacturing of such devices.”

1. A. Markov, H. Guerboukha, and M. Skorobogatiy, “Hybrid metal wire–dielectric terahertz waveguides: challenges and opportunities,” *JOSA B* **31**(11), 2587-2600 (2014).
2. M. K. Mridha, A. Mazhorova, M. Clerici, I. Al-Naib, M. Daneau, X. Ropagnol, M. Peccianti, C. Reimer, M. Ferrara, L. Razzari, and F. Vidal, “Active terahertz two-wire waveguides,” *Opt. Express* **22**(19), 22340-22348 (2014).
3. G. Yan, A. Markov, Y. Chinifooroshan, S. M. Tripathi, W. J. Bock, and M. Skorobogatiy, “Low-loss terahertz waveguide Bragg grating using a two-wire waveguide and a paper grating,” *Opt. Lett.* **38**(16), 3089-3092 (2013).
4. Y. Cao, K. Nallappan, H. Guerboukha, G. Xu, and M. Skorobogatiy, “Additive manufacturing of highly reconfigurable plasmonic circuits for terahertz communications,” *Optica* **7**(9), 1112-1125 (2020).
5. S. Jia, X. Pang, O. Ozolins, X. Yu, H. Hu, J. Yu, P. Guan, F. Da. Ros, S. Popov, G. Jacobsen, and M. Galili, “0.4 THz photonic-wireless link with 106 Gb/s single channel bitrate,” *J. Light. Technol.* **36**(2), 610-616 (2018).
6. S. Jia, M. C. Lo, L. Zhang, O. Ozolins, A. Udalcovs, D. Kong, X. Pang, X. Yu, S. Xiao, S. Popov, J. Chen, G. Carpintero, T. Morioka, H. Hu, and L. K. Oxenlewe, “Integrated dual-DFB laser for 408 GHz carrier generation enabling 131 Gbit/s wireless transmission over 10.7 meters,” in *Optical Fiber Communication Conference* (2019), Th1C-2.
7. K. Liu, S. Jia, S. Wang, X. Pang, W. Li, S. Zheng, H. Chi, X. Jin, X. Zhang, and X. Yu, “100 Gbit/s THz photonic wireless transmission in the 350-GHz band with extended reach,” *IEEE Photon. Technol. Lett.* **30**(11), 1064-1067 (2018).
8. J. Ma, N. J. Karl, S. Bretin, G. Ducournau, and D. M. Mittleman, “Frequency-division multiplexer and demultiplexer for terahertz wireless links,” *Nat. Commun.* **8**(1), 1-8 (2017).
9. N. J. Karl, R. W. McKinney, Y. Monnai, R. Mendis, and D. M. Mittleman, “Frequency-division multiplexing in the terahertz range using a leaky-wave antenna,” *Nat. Photonics* **9**(11), 717-720 (2015).
10. M. Yata, M. Fujita, and T. Nagatsuma, “Photonic-crystal diplexers for terahertz-wave applications,” *Opt. Express* **24**(7), 7835-7849 (2016).
11. K. S. Reichel, N. Lozada-Smith, I. D. Joshipura, J. Ma, R. Shrestha, R. Mendis, M. D. Dickey, and D. M. Mittleman, “Electrically reconfigurable terahertz signal processing devices using liquid metal components,” *Nat. Commun.* **9**(1), 1-6 (2018).
12. D. Headland, W. Withayachumnankul, M. Fujita, and T. Nagatsuma, “Gratingless integrated tunneling multiplexer for terahertz waves,” *Optica* **8**(5), 621-629 (2021).

Query 3. In the communications experiment that is presented in Fig. 6, a clear channel is not obtained for the add and drop signal paths, which exhibit BER>10⁻⁵ even for a relatively low data rate of 1 Gbit/s. This is a thousand times more bit errors than for the through channel. Is there some additional dispersion or reflection in the add/drop paths?

Q3. Author Response:

First, we would like to address the reason for the low SNR of the received THz signal. The ADM was characterized using an in-house photonic-based THz communication system having the output power of THz emitter (UTC-PD) in the range between 125 μ W (-9 dBm) and 250 μ W (-6 dBm) within the spectral range of 120-160 GHz. This is the only system available in the Montréal region to perform BER testing at THz frequencies. The target BER was set to 10^{-12} . Due to the low output power of the employed THz emitter (photomixer), high insertion losses of the ADM (~11 dB for the channel Through and > 14 dB for the channel Add/Drop actions respectively), as well as the coupling loss between ADM and THz emitter and detector (~3 dB at each interface), the power of THz signal received by the THz detector is far below the limit for error-free detection, which is ~ -20 dBm for operation around 140 GHz [2]. Therefore, the error-free data transmission cannot be currently realized for any of the ADM channel actions with the available equipment. Nevertheless, the high value of BER obtained with an optics-based THz communication system does not mean that the clear channel cannot be achieved with the demonstrated ADM circuits. To demonstrate a clear channel we will simply need to employ a higher power THz emitter (a few dBm output) which currently become more widely available in the electronic-based systems [3,4], while we will also need to improve the performance of the currently employed Y-splitter. While important, these requirements for a clear channel demonstration do not pose fundamental barriers for the technology described in this work, neither they diminish its impact.

Now, we would like to address the issue of why BER for the Add/Drop signal is much higher than the one for the channel Through one. This is mainly attributed to the higher insertion losses (> 14 dB) for the Add/Drop signal propagating within the grating stopband (~150 GHz) compared to the lower insertion losses (~11 dB) for the signal propagating from the In to Through port within the grating passband (~130 GHz) of the complete ADM circuit when it is integrated into two-wire THz waveguide-based communication networks, see Figs. S11 and S14 for details. Comparing eye diagrams of the THz signals at 1 Gbps propagating through the ADM with balanced Add/Drop action with that of the noise level (see Figs. 6(b) and S15(e)), the SNR for channel Drop and Through actions are estimated at 5 dB and 11 dB, respectively. Considering the exponential relation between BER and SNR ($BER \sim \exp(-SNR)$), it is not surprising to see that the BER for the channel Drop action is a thousand times larger than that for the channel Through, which is in good agreement with the experimental result shown in Fig. 6(c).

Moreover, as rightfully noted by the reviewer, waveguide dispersion and parasitic reflections in the ADM circuits can further deteriorate the ADM performance, and Supplementary Note 12 is dedicated to this issue. Particularly, the bent two-wire waveguide features significantly higher GVD (~ 10 ps/(THz*cm)) than the straight one (~ 3 ps/(THz*cm)) at ~ 150 GHz [5]. For the channel Through action, the signal within the grating passband propagates along a straight path from In port, while for the channel Add/Drop actions the signal within the grating stopband propagates along the bent section of a Y-splitter. As a result, higher dispersion of the curved waveguides (see Eq. S3) sets a lower limit on the maximal bitrate supported by the Add/Drop action (28 Gbps) compared to the one for the Through action (40 Gbps). More importantly, the channel bandwidth for the Add/Drop action is further restricted to ~ 2 -4 GHz due to fracturing of the grating stopband because of the parasitic interference effects within the ADM circuit. In contrast, the grating passband features a much larger bandwidth, thus resulting in a better performing Through action.

All these factors contribute to a stronger performance of the Through action compared to the Add/Drop actions.

Q3. Author Action:

The following paragraphs are added to the main text of the paper and to Supplementary Note 11 to clarify the key challenges that impede error-free data transmission in the ADM circuits demonstrated in this paper, as well as to explain the observed difference in the BER response of the channel Add/Drop and Through actions.

A3. At the end of Chapter 2.4 of the revised manuscript we write:

“Losses of various ADM subcomponents (quoted by power in the following discussion) affect significantly and unequally the BER of the Add/Drop and Through actions due to the negative effect of the noise on the detection accuracy. The highest loss in our device ~ 8 dB is a bending (radiation) loss of the 6.3cm-long curved branch of a Y-splitter that mostly affects the Add/Drop action. The grating-loaded side coupler section results in ~ 7 dB loss for operation in the grating passband (Through action), while resulting in over 4 dB loss for operation in the grating stopband (Add/Drop actions). These losses are due to several factors including ohmic losses, the mismatch between the modal fields of a straight waveguide and a grating, as well as scattering on the imperfections in the grating structure. Next, there is butt-coupling loss of ~ 3 dB at each interface between ADM and the waveguide-coupled THz emitter and detector [29]. Finally, there are relatively low losses of straight waveguides (0.4 dB/cm), which are mostly due to ohmic losses incurred by the plasmonic modes in the silver layers. As a result, the relatively high insertion losses of the developed ADMs (~ 11 dB for the channel Through and > 14 dB for the channel Add/Drop actions) and the coupling loss between ADM and THz emitter and detector (~ 3 dB at each interface), together with the low power of the employed THz emitter (-9 dBm – -6 dBm in the 120-160 GHz spectral range) exceed the limit for the error-free data transmission, which for our THz communication system is ~ 20 dBm [55]. This results in the relatively low signal-to-noise ratio during BER measurements (see Supplementary Note 11) and, as a consequence, high and unequal BERs for the channel Add/Drop and Through actions.

*It is also important to comment on the potential effect of the group velocity dispersion on the ADM performance (see Supplementary Note 12 for details) as strong modal dispersion can lead to signal scrambling that manifests itself in the eye diagram distortion and closing of the eye pattern. In our case, however, both straight and bent waveguides are short (~ 10 cm) and feature relatively small dispersions (~ 3 and 10 ps/(THz*cm), respectively), therefore, the impact of modal dispersion on the quality of transmitted data is relatively small compared to the effect of ADM losses discussed earlier. A detailed analysis shows that, ultimately, the Through action is indeed limited by the dispersion of a straight waveguide, with maximal bitrates in the 10s of Gbps. At the same time, the Add/Drop action is rather limited by the size and shape of the grating stopband, which in our case limits the bitrates to ~ 4 -8 Gbps.”*

29. Y. Cao, K. Nallappan, H. Guerboukha, G. Xu, and M. Skorobogatiy, “Additive manufacturing of highly reconfigurable plasmonic circuits for terahertz communications,” *Optica* 7(9), 1112-1125 (2020).

55. K. Nallappan, Y. Cao, G. Xu, H. Guerboukha, C. Nerguizian, and M. Skorobogatiy, “Dispersion-limited versus power-limited terahertz communication links using solid core subwavelength dielectric fibers,”

B3. The difference in the BERs for the channel Add/Drop and Through actions are also addressed in Supplementary Note 11:

“.....The low value of SNR results in the failure of the error-free data transmission when using a photonic-based THz communication system for ADM characterization. As the BER depends exponentially on SNR ($BER \sim \exp(-SNR)$), and insertion losses of the Add/Drop action are significantly higher than those of a Through action, it is expected that the BER for channel Add/Drop to be much higher than that for the channel Through. In fact, comparing eye diagrams of the THz signals at 1 Gbps

propagating through the ADM (see Fig. 6(b)) with that of the noise level (see S15(e)), the SNR for channel Drop and Through actions are estimated at ~5 dB and ~11 dB respectively. From this, we expect that BER for the Add/Drop action should be several thousand times higher than that for the Through action, which is in good agreement with the experimental result shown in Fig. 6.”

Query 4. The authors have employed Y-junctions to interface the individual ports with the Bragg mirror. However, an ideal Y-junction exhibits 3 dB of loss when splitting, and also when joining, due to reciprocity. All signal paths through this device (i.e. add, drop, and through) must pass through two Y-junctions, for a minimum of 6 dB loss, or a maximum efficiency of 25%. Can the authors justify this innately low efficiency?

Q4. Author Response:

The reviewer is certainly correct by stating that a classical Y-splitter should induce a 3 dB insertion loss. In fact, in our work, we use both a Y-splitter and a Y-coupler. While the two devices sound the same their function and operation are completely different, which, we believe, can confuse the reviewer and the readers. In fact, a Y-coupler (see Fig. S6 (a)) features two physically coalescing waveguide bends. Thus, when launching light in port 3, it will be equally split into port 1 and port 2. The Y-coupler was used in the measurements of the WBG relative reflection coefficient (see Fig. S12(b)). There, the light was launched into port 1, collected at port 2, while the Y-coupler, indeed brought an additional 3 dB loss due to reflection of the THz signal either from the WBG or the reference mirror at port 3 [5], thus resulting in the low SNR of the measurement.

In contrast, the Y-splitter used in the ADM circuits for beam steering (see Figs. 1 and 3(b)) is different from the Y-coupler both in geometry and function. As seen from Fig. S6(b), the two waveguides in the splitter (one straight, another bent) never physically coalesce. In fact, the Y-splitter is designed so that its cross-section at port 3 is identical to that of a directional side coupler used in the ADM. As a result, at port 3 there is not one, but two physically distinct waveguides (unlike in the case of a port 3 of a Y-coupler), and there is no additional 3 dB loss incurred at port 3 of a Y-splitter. The leading loss mechanism of an ADM Y-splitter is related to the propagation losses of a straight and a curved waveguide. Finally, due to mismatch in the modal field distributions of the straight and curved waveguides of otherwise identical cross-sections, one also expects an additional small insertion loss at the coupling plane between a side coupler and a Y-splitter (of identical cross-sections).

Fig. S6 Schematics of (a) Y-coupler and (b) Y-splitter (top part removed). The red dotted regions show Y junctions in each component.

Q4. Author Action:

A4. To clarify that there is no 3 dB butt-coupling loss at the interface of a Y-splitter and a side coupler section, we added the following conclusion at the end of the second paragraph of the revised Chapter 2.2

“The Y-splitter features a curved two-wire waveguide gradually approaching a straight one in a wedge configuration (the two waveguides do not coalesce). The seamless connection between a Y-splitter and a side coupler section is assured as their cross-sections at the coupling planes (Planes 1 and 2) are the same by design. Therefore, butt-coupling of these components brings only a small insertion loss due to mismatch of the modal fields of straight and bent waveguides with otherwise identical cross-sections. We also note that Y-splitters used in the ADM have to be distinguished from the Y-couplers used for characterization of the WBG reflection coefficient (see discussion in Supplementary Note 5), as the latter usually bring additional 3 dB loss.”

B4. To explain the difference between a Y-coupler and a Y-splitter we placed Fig. S6 and an explanatory note in Supplementary Note 5 entitled “Y-splitters used in the directional coupler and ADM circuits”:

“In our work, we use both a Y-splitter and a Y-coupler. While the two devices sound the same their function and operation are different. Thus, a Y-coupler (see Fig. S6 (a)) features two physically coalescing bent waveguides. Thus, when launching light into port 3, it will be equally split into port 1 and port 2. In this work, the Y-coupler is used to measure the WBG relative reflection coefficient (see Supplementary Note 9). There, the light is launched into port 1 and collected at port 2, while the THz signal is back-reflected at port 3 either from the WBG or the reference mirror [4]. Note that Y-couplers bring an additional 3 dB loss after reflection at port 3 and equal splitting into ports 1 and 2, thus lowering the SNR of the measurement.

In contrast, a Y-splitter used in the ADM circuits for beam steering (see Figs. 1 and 3(b)) is different from the Y-coupler both in geometry and in function. As seen from Fig. S6 (b), the two waveguides in the splitter (one straight, another bent) never physically coalesce. In fact, the Y-splitter is designed so that its cross-section at port 3 is identical to that of a directional coupler of the ADM circuit. As a result, at port 3 there is not one, but two physically distinct waveguides (unlike in the case of a single waveguide at port 3 of a Y-coupler), and there is no additional 3 dB loss incurred at port 3 of a Y-splitter. In fact, the leading loss mechanism of a Y-splitter is simply related to the propagation losses of the straight and curved waveguides of the two branches. Finally, we note that due to mismatch in the modal field distributions of the straight and curved waveguides of otherwise identical cross-sections, one also expects an additional small insertion loss at the coupling plane between a side coupler and a Y-splitter (of identical cross-sections).”

4. Y. Cao, K. Nallappan, H. Guerboukha, G. Xu, and M. Skorobogatiy, “Additive manufacturing of highly reconfigurable plasmonic circuits for terahertz communications,” *Optica* 7(9), 1112-1125 (2020).

Query 5. What is the size of the overall device? And how does it compare to contemporary MMIC systems?

Q5. Author Response:

The two-wire ADMs with a size of 12.5 cm by 5 cm were developed using the micro-encapsulated two-wire waveguides with a cross-sectional area of $\sim 1.3 \text{ cm}^2$ (including the cage). A straight waveguide bridging the In and Through ports has a total length of 12.5 cm and is made of three sections (straight 4.5 cm-long branches of the two Y-splitters and a 3.5 cm-long grating-loaded side coupler). The Y-splitters also comprise a 90° waveguide bend of 4 cm bending radius.

In contrast, the MMICs for operation in the THz regime, where various active and passive components are integrated into a planar wafer in compound semiconductor, usually feature an area of few square millimeters and a thickness of tens of micrometers [6-8], which is much more compact than the proposed three-dimensional two-wire waveguide-based THz plasmonic circuits. Nevertheless, the proposed two-wire circuits outperform the MMICs in several other aspects such as the threshold for entering into production, reconfigurability, the possibility of tuning, the ability of 3D integration, as well as operation without an external power supply.

Q5. Author Action:

A5. The length of each waveguide section of the ADM circuit has been indicated in Fig. 1.

B5. The comparison between MMIC and our proposed two-wire THz plasmonic circuit has been added as the eighth paragraph in the Introduction Chapter, which is as follow,

“Finally, it is important to mention the versatile monolithic microwave integrated circuits (MMIC) that are built on planar semiconductor wafers, and whose application has extended into the THz field [38-41]. In comparison, while our plasmonic waveguide circuits are less compact than MMICs in the 2D plane, however, they carry several other advantages including high reconfigurability and tunability, ease of manufacturing and packaging, as well as the ability of native 3D integration. Furthermore, different from the MMICs whose operation requires external supply power, signal processing in the two-wire plasmonic circuits is completely passive and more reliable when operating in complex electromagnetic environments.”

38. I. Kallfass, I. Dan, S. Rey, P. Harati, J. Antes, A. Tessmann, S. Wagner, M. Kuri, R. Weber, H. Massler, A. Leuther, T. Merkle, and T. Kurner, “Towards MMIC-based 300GHz indoor wireless communication systems,” *IEICE Trans Inf Syst* 98(12), 1081-1090 (2015).

39. W. Deal, X. B. Mei, K. M. Leong, V. Radisic, S. Sarkozy, and R. Lai, “THz monolithic integrated circuits using InP high electron mobility transistors,” *IEEE Trans. Terahertz Sci. Technol.* 1(1), 25-32 (2011).

40. H. J. Song, J. Y. Kim, K. Ajito, N. Kukutsu, and M. Yaita, “50-Gb/s direct conversion QPSK modulator and demodulator MMICs for terahertz communications at 300 GHz,” *IEEE Trans. Microw. Theory Techn.* 62(3), 600-609 (2014).

41. W. R. Deal, K. Leong, A. Zamora, V. Radisic, and X. B. Mei, “Recent progress in scaling InP HEMT TMIC technology to 850 GHz,” in *IEEE MTT-S International Microwave Symposium* (2014), 1-3.

C5. A more detailed discussion of the advantages and potential applications of the proposed two-wire THz plasmonic circuits has also been given in the Introduction Chapter as well as in the Supplementary Note 1 entitled “Further discussion of the novelty and key contributions of this work”. (See more details in response to your Query #2).

Query 6. Is this device compatible with standard mm-wave connectors? If so, why have the authors chosen to employ a lens-coupled photomixer that is intended to interface with free space? This choice must be justified, as the authors directly attribute spurious features of the measured results to this photomixer in lines 157-159: “...strong ripples that are presented in the experimental reflection spectra (see Fig. 2(d)). These result from the standing waves formed in the cavity of the CW-THz spectroscopy setup, as well as inside of a photomixer silicon lens.”?

Q6. Author Response:

The ADM circuit is built on a micro-encapsulated two-wire THz plasmonic waveguide whose design has been optimized to butt-couple the standard WR6.5 rectangular waveguide. The power coupling coefficient between the TEM mode supported by this two-wire waveguide and the TE₁₀ mode supported by the WR6.5 rectangular waveguide reaches ~48% at 140 GHz. The high excitation efficiency can be maintained with only ~6% variation across the operational frequency range of the rectangular waveguide (110 GHz-170 GHz). Additionally, for the convenience of connectorization with the WR6.5 rectangular waveguide, an alignment feature (screw holes matching the UG-387/U-M flange) is integrated via 3D printing onto both end facets of the two-wire waveguide components [5]. Therefore, the proposed two-wire THz plasmonic circuit is perfectly suitable for characterization by connectorizing it to various THz emitter/detector modules via standard hollow rectangular waveguides.

Originally, we characterized the two-wire waveguide components and circuits using an optics-based free-space CW-THz spectroscopy system and its THz communications derivative which were built and currently used in our research group for a variety of projects in spectroscopy, sensing, and communications. These systems have an ultra-broad frequency operation range (from 80 GHz to 1.2 THz), an outstanding spectral resolution (~10 MHz), and are easily tunable in the whole spectral range. One of the key disadvantages of such systems is a relatively lower power compared to their solid-state counterparts. A key element of an optics-based CW-THz spectroscopy system is an InGaAs photomixer that features an integrated silicon lens to focus a free space THz beam onto a chip. Unfortunately, multiple reflections inside a silicon lens result in intensity ripples with a frequency of ~4 GHz, which are superimposed onto the measured transmission spectra [9]. Combined with the low power of the generated THz radiation (less than -10 dBm within 120-160 GHz), it results in noisy spectra with some spurious oscillations even after normalization.

Fig. S4. Schematic of the network analyzer to characterize THz two-wire waveguide components. The red dotted box shows the connection between two-wire waveguide components and extender modules via rectangular waveguides.

In the revised work, to increase the quality of optical characterization and reduce the effect of noise on the system performance, we redid the optical characterization of all the THz components and circuits using a network analyzer (N5247B PNA-X, Keysight) and solid-state THz sources (Vector Network Analyzer Extenders, Virginia Diodes) that belong to another research center in the Montréal region. Since no standing wave is formed inside the THz emitter and detector modules, the reference transmission spectrum, which was obtained by connecting extenders via a rectangular waveguide, is relatively smooth within the studied frequency range (120-160 GHz). To characterize the two-wire waveguide components and circuits, they were placed inside the spectroscopy system and butt-coupled to the rectangular waveguides connecting the THz extenders (see Fig. S4). However, even in this configuration, there are still some standing waves that are excited in the rectangular waveguides due to impedance mismatch between the rectangular waveguide and the two-wire waveguides used in the ADMs and components. While this mismatch still results in oscillations in the recorded transmission and reflection spectra, however, due to the high SNR enabled by the high output power of the THz sources, such spectral oscillations can be practically removed via normalization to a properly

chosen reference. Therefore, the new experimental results (see Figs. 2 and 5) reflect better the actual spectral response of the studied two-wire waveguide components and show much fewer spurious features than prior measurements with an optics-based system.

Q6. Author Action:

A6. We have revised the fifth paragraph of the Introduction Chapter to emphasize the compatibility of the proposed two-wire waveguide with the standard mm-wave connectors (rectangular waveguide).

“...Resultant two-wire waveguides feature some of the lowest transmission and bending losses, very low group velocity dispersion (GVD), and broadband operation compared to other THz waveguides [31,32]. Such waveguides can be also easily and with high coupling efficiency butt-coupled to rectangular waveguides that are prevalent in the microwave and THz communications [29].”

29. Y. Cao, K. Nallappan, H. Guerboukha, G. Xu, and M. Skorobogatiy, “Additive manufacturing of highly reconfigurable plasmonic circuits for terahertz communications,” *Optica* 7(9), 1112-1125 (2020).

31. M. Mbonye, R. Mendis, and D. M. Mittleman, “A terahertz two-wire waveguide with low bending loss,” *Appl. Phys. Lett.* 95(23), 233506 (2009).

32. K. Wang, and D. M. Mittleman, “Metal wires for terahertz wave guiding,” *Nature* 432(7015), 376-379 (2004).

B6. In the Methodology section, we replaced the description of a CW-THz spectroscopy system with the one based on the network analyzer. We have also updated the experimental procedures that we used to characterize the two-wire components and circuits using the network analyzer.

“The two-wire waveguide components and circuits were characterized using a network analyzer (N5247B PNA-X, Keysight) together with the frequency multiplier modules (Vector Network Analyzer WR5.1 and WR8.0 Extenders, Virginia Diodes).

The WBGs were characterized as follows. Firstly, a two-wire waveguide section featuring the same 2.5 cm length as the WBGs was placed between the two two-wire waveguide sections (see Fig. S4). This assembly was then butt-coupled on both sides to the rectangular waveguides of the two extender modules. Then, the $S_{r,21}^2$ (transmission by power) of the assembly was measured as a reference. Next, the 2.5cm-long waveguide section was replaced by the WBG element, and the S_{21}^2 and S_{11}^2 of the assembled waveguide component were recorded. The relative transmission coefficient of the WBG was obtained by comparing the measured S_{21}^2 value with that of the reference. Finally, the WBG element was removed, and a planar metallic mirror was placed in its place, while the $S_{r,11}^2$ was remeasured to serve as a reference for the WBG reflection coefficient estimation. To remove oscillations from the reflected spectra, we used the midpoint between the adjacent maxima and minima of the recorded reference $S_{r,11}$ to fit the reference spectra with a smooth curve. Then, we normalized the WBG reflection coefficient by dividing the midpoint between the adjacent maxima and minima on the recorded WBG spectra S_{11} by the fitted reference to obtain the data shown in Fig. 2 (d) (See Supplementary Note 4 for more details).”

C6. The experimental results obtained by the network analyzer are now shown in the main text (see Figs. 2 and 5 in the Author Action to your Query #1), while the ones previously measured by the CW-THz spectroscopy system have been moved to Supplementary Note 9 entitled “*Characterization of two-wire waveguide-based Bragg gratings and ADM circuits using free-space continuous-wave (CW) THz spectroscopy system*”. At the same time, the BER measurements were still performed using an optics-based system as we could not find any solid-state system in the Montreal area that could perform such measurement at ~150 GHz.

D6. Based on the new experimental results, the analysis of the two-wire WBGs and grating-loaded side couplers have been revised (see the Author action to your Query #1)

Query 7. This is a minor issue, but the quality of English language could be improved. For instance, there are errors in the abstract: “Terahertz band is considered as...” -> “The terahertz band is considered to be...”, and “...attention is payed...” -> “...attention is paid...”. I recommend having a native speaker of the English language review the manuscript.

Q7. Author Response:

We have carefully proofread the main text and supplementary notes to correct errors with the assistance of professional native speakers.

Reviewer 2:

First, we would like to thank the reviewer for their time and thoughtful comments related to our work. To address the reviewer's concerns, we revised the figures and text to make the main text concise and easier to understand, while we also added additional important information in the supplementary notes. Here is our detailed reply:

For convenience, we indicate in **bold** the original questions by the reviewer, in regular font our response to the reviewer, and in *italic* our corrections to the manuscript.

This paper shows fabrication and characterization of a 3d printed add-drop multiplexer for THz communications using fiber Bragg grating. This subject is of relevance since the THz band is the future of wireless communications. To reach the goal of truly functioning THz systems, signal processing devices are required. Here, the researchers demonstrate a 3d printed add-drop multiplexer. This work is novel in the strategy used and can definitely attract the interest of the scientific community working in wireless communications at high frequencies. The design strategy is well documented with both simulations and experiments, which are carried using both a THz spectrometer and a communications system. The conclusions are well supported, and all the relevant details are provided in the supplementary (which is something that is not usually seen with such detail). I have however multiple comments about the general presentation. Reading the manuscript is tedious and some of the figures are hard to understand (see detailed comments below). Based on these comments, I believe this paper can be accepted if the researchers correct the following points, which I consider minor because they are mainly related to the format:

The following are our replies to the reviewer queries:

Query 1. Abstract. Line 13: payed -> paid. Line 16: that THE proposed

Q1. Author Response:

We have carefully proofread the main text and supplementary notes to correct grammatical errors.

Query 2. In the introduction, optical networks are compared to terahertz networks. However, the mentioned optical networks operate in fiber, whereas the comparison should also be done with microwave/RF WIRELESS communications.

Q2. Author Response:

This is indeed a valid and interesting point. In fact, as the THz spectrum finds itself between microwave and optical domains, many studies in THz communications adapt the components and system-design concepts developed for either near-infrared or microwave. Moreover, in the THz frequency range, both fiber and wireless links are equally possible with already available hardware, which opens an interesting opportunity for hybrid networks and novel data transmission modalities. It is, thus, important to mention that Frequency Division Modality that we study in this paper is already implemented in both optics and microwave/RF wireless links, therefore FDM implementation in the THz frequency range is natural and viable. Particularly, in the microwave region, we note the orthogonal frequency division multiple access technology, which is an FDM derivative implemented in the latest 4G LTE and 5G mobile communication networks.

Q2. Author Action:

A2. The FDM application in the microwave regime is now discussed at the end of the first paragraph in the Introduction Chapter: *“Among them, frequency division multiplexing (FDM), in which several discrete carrier frequencies support distinct users, is routinely used in fiber optic and wireless communications to multiply the data-throughput capacity. In optical networks, one typically uses carrier waves in the C-band (infrared) with channel spacing less than 10 GHz, thus allowing simultaneous transmission of hundreds of channels with a hundred MHz bandwidth each [13]. Furthermore, in the microwave/RF domain, the*

FDM technology is also prevalent in the IEEE 802.11 wireless and mobile networks. Particularly, in 4G LTE and 5G cellular communication networks, the orthogonal frequency division multiple access is implemented using electronic circuits for FFT and IFFT, where thousands of tightly packed sub-carriers within tens of MHz bandwidth are spaced by tens of kHz [14]. As THz frequencies find themselves between infrared and microwave bands in the electromagnetic spectrum, the THz FDM communications can benefit from both the optical and electronic solutions. The lower THz band 100 – 300 GHz, in particular, can accommodate multiple channels with bandwidths from several GHz to several 10s of GHz, thus, allowing spectral allocation and simultaneous utilization of numerous high-bandwidth channels both within the THz fiber-based and THz wireless communication modalities.”

13. H. Toba, K. Inoue, and K. Nosu, "A conceptual design on optical frequency-division-multiplexing distribution systems with optical tunable filters," IEEE J. Sel. Areas Commun. 4(9), 1458-1467 (1986).

14. H. Yin, and S. Alamouti, "OFDMA: A broadband wireless access technology," In 2006 IEEE sarnoff symposium (1-4). IEEE. (2006, March).

Query 3. Line 70: drop/add -> add/drop

Q3. Author Response:

We have carefully proofread the main text and supplementary notes to correct grammatical errors.

Query 4. It is mentioned that thermal and mechanical tuning of the proposed circuits is possible. First, how? Second, thermal and mechanical tuning are not preferred for dynamic band allocation. How are the proposed circuits better than the ones based on a silicon platform, since these can be modulated with charge injection? How something similar can be done with a purely metallic waveguide?

Q4. Author Response:

Indeed, this is a valid and important question. First, we note that by dynamic bandwidth allocation we mean the ability to change the ADM frequency response (such as the value of the add/drop frequency) via the application of some external stimuli such as temperature, mechanical stress, etc. Naturally, circuit switching time will depend crucially on the tuning mechanism. For thermal tuning, the switching time is typically in seconds, while for mechanical tuning it can be in milliseconds. If faster switching times are needed, one normally resorts to optical or electric tuning. That said, even the circuits with second-long switching times can find their applications in different parts of the network.

Considering that the two-wire waveguide-based ADM circuits presented in this work are built on a 3D printed resin substrate, the thermal and mechanical tunability is readily achievable by changing the physical properties and geometry of the resin substrate. In fact, in the newly added Supplementary Note 3 (see also Author Action B4), we have experimentally verified the feasibility of thermal tuning of the Waveguide Bragg Grating spectral response by placing it on top of a heating element. While the further investigation of tuning is in order, this experiment demonstrates the feasibility of such a tuning. Furthermore, mechanical tuning is possible by using the grating-loaded side coupler built on compressible dielectric support. Under axial strain, the deformation would result in the change of a grating period, and as a consequence, a shift in the Add/Drop frequency. Practically, this can be realized by using a variety of commercially available soft photosensitive resins when 3D printing the ADM structure.

That said, we agree with the reviewer that slow response of the thermo-mechanical actuation impedes their application for true dynamic bandwidth allocation, while it is certainly suitable for applications in quasi-static network switching and bandwidth allocation. In this respect, our metallic waveguides in their current form cannot compete with a silicon platform that profits from ultra-fast charge injection tuning.

Nevertheless, several routes can be pursued to make charge injection tuning possible in the case of two-wire waveguides. One is deposition on top of metallic layers of a semiconductor material (ex. InSb, etc.) that features plasma frequency in the near THz frequency range [10]. In such materials, the plasmonic frequency can be readily changed via carrier injection, thus altering the optical response of a waveguide. Another is the insertion of an optically or electrically tunable film directly into the air gap of a two-wire waveguide. By changing the conductivity of such a film, one can realize phase and amplitude modulation of the transmitted THz mode. Particularly, in the case of graphene sheets, their complex conductivity can be adjusted on a timescale of picoseconds by changing its Fermi energy via optical pumping or electrical gating [11-13], while their ability to modulate THz wave propagating in a two-wire waveguide was recently demonstrated in [14,15]. We, therefore, believe that the integration of

active materials into the two-waveguide structure can make such waveguides tunable both optically and electrically. Finally, in contrast with a silicon platform which features a fixed planar structure and tightly confined modal fields, the air core two-wire waveguides feature a highly flexible 3D structure and easily accessible modal fields, thus allowing a variety of tuning modalities ranging from the relatively slow thermal and mechanical actuation to ultra-fast optical and electronic tuning.

Q4. Author Action:

A4. We believe that a detailed study of the tunable THz plasmonic circuits and their viability for THz communications merits a separate paper. Nevertheless, in this work we still want to indicate this possibility in the Introduction Chapter (seventh paragraph): *“Furthermore, thermal and mechanical tuning of the proposed circuits is readily achievable by changing the physical properties or geometry of the resin substrate and metallic film (see Supplementary Note 3 for an example). As the air-core two-wire waveguides feature a highly flexible structure and easily accessible modal fields, various elements can be inserted into the air gap to modify or tune the waveguide propagation properties. Moreover, optical and electrical tuning of such waveguides is also possible as was recently demonstrated via insertion of a graphene sheet between the two wires, which was then tuned via optical pumping or electrical gating [33-36]. Such structures promise solutions for the important problem of dynamic band allocation in the THz communication networks [37].”*

33. M. Chen, P. Sheng, W. Sun, and J. Cai, "A symmetric terahertz graphene-based hybrid plasmonic waveguide," *Opt. Commun.* **376**, 41-46 (2016).

34. W. Xu, Z. H. Zhu, K. Liu, J. F. Zhang, X. D. Yuan, Q. S. Lu, and S. Q. Qin, "Toward integrated electrically controllable directional coupling based on dielectric loaded graphene plasmonic waveguide," *Opt. Lett.* **40**(7), 1603-1606 (2015).

35. Z. Wang, J. Qiao, S. Zhao, S. Wang, C. He, X. Tao, and S. Wang, "Recent progress in terahertz modulation using photonic structures based on two-dimensional materials," *InfoMat* **3**(10), 1110-1133 (2021).

36. M. Liu, X. Yin, E. Ulin-Avila, B. Geng, T. Zentgraf, L. Ju, F. Wang, and X. Zhang, "A graphene-based broadband optical modulator," *Nature* **474**(7349), 64-67 (2011).

37. A. Sabharwal, A. Khoshnevis, and E. Knightly, "Opportunistic spectral usage: Bounds and a multi-band CSMA/CA protocol," *IEEE/ACM Trans. Netw.* **15**(3), 533-545 (2007).

B4. We added Supplementary Note 3 entitled *“Tunability of 3D printed two-wire plasmonic waveguide components”*, where the spectral performance of a two-wire WBG under different operating temperatures was studied to demonstrate the feasibility of thermal tuning of the ADM circuits. For completeness, in the same note we present a discussion of various possible tunability modalities as mentioned in the reviewer response above:

Fig S2. The relative transmission coefficient (by field) of a two-wire WBG (20 periods, $\Lambda=1.03$ mm, $H=0.21$ mm) operating under different temperatures.

“To demonstrate the potential for thermal tunability of the 3D printed two-wire waveguide components presented in this work, we experimentally study the change in the spectral response of a WBG under different operating temperatures (see Fig. S2). The WBG was placed on top of a PTC heater with a temperature-controlled by the applied voltage. In our experiments, the operating temperature of a WBG reached 70 °C with the applied voltage of 12 V. It is found that at higher operating temperatures, the spectral position of the WBG stopband shifts towards lower frequencies. This is attributed to the expansion of the plastic material under heating, and as a consequence, an increase in the grating period. As WBG is a principal frequency-selective element in the ADM structure presented in this work, thermal tuning of the WBG response opens the way for thermal tuning of the complete ADM circuit.

More generally, tunable ADMs can be of great benefit for applications in dynamic bandwidth allocation, which, in practical terms means the ability to change the ADM frequency response (such as the value of the Add/Drop frequency) via the application of some external stimuli such as temperature, mechanical stress, etc. The circuit switching time will depend crucially on the tuning mechanism. For thermal tuning, the switching time is typically in seconds, while for mechanical tuning it can be in milliseconds. If faster switching times are needed, one normally resorts to optical or electric tuning.

Considering that the two-wire waveguide-based ADM circuits presented in this work are built on a 3D printed resin substrate, the thermal and mechanical tunability is readily achievable by changing the physical properties and geometry of the resin substrate. Thermal tuning is normally achieved by changing the temperature of the frequency-selective element of an ADM such as the WBG used in the example above. Furthermore, mechanical tuning of an ADM presented in this work is possible by using the grating-loaded side coupler built on a compressible dielectric support. Under axial strain, the deformation would result in the change of a grating period, and as a consequence, a shift in the Add/Drop frequency. Practically, this can be realized by using a variety of commercially available soft photosensitive resins when 3D printing the ADM structure.

That said, the slow response of the thermo-mechanical actuation impedes its application for true dynamic bandwidth allocation, while it is certainly suitable for applications in quasi-static switching. In this respect, the classic two-wire metallic waveguides cannot compete with a silicon platform that profits from the possibility of ultra-fast charge injection tuning. That said, several routes can be pursued to make charge injection tuning possible in the case of two-wire waveguides. One is deposition on top of metallic layers of a semiconductor material (ex. InSb, etc.) that features plasma frequency in the near THz frequency range [13]. In such materials, the plasmonic frequency can be readily changed via carrier injection, thus altering the optical response of a waveguide. Another is the insertion of an optically or electrically tunable film directly into the air gap of a two-wire waveguide. By changing the conductivity of such a film, one can realize phase and amplitude modulation of the transmitted THz mode. Particularly, in the case of graphene sheets, their complex conductivity can be adjusted on a timescale of picoseconds by changing its Fermi energy via optical pumping or electrical gating [14-16], while their ability to modulate THz wave propagating in a two-wire waveguide was recently demonstrated in [17,18]. We, therefore, believe that the integration of active materials into the two-waveguide structure can make such waveguides tunable both optically and electrically.

Therefore, we believe that in contrast with a silicon platform that features a fixed planar structure and tightly confined modal fields, the air core two-wire waveguides feature a highly flexible 3D structure and easily accessible modal fields, thus allowing a variety of tuning modalities ranging from the relatively slow thermal and mechanical actuation to ultra-fast optical and electronic tuning.”

13. M. Poulin, S. Giannacopoulos, and M. Skorobogatiy, “Surface Wave Enhanced Sensing in the Terahertz Spectral Range: Modalities, Materials, and Perspectives,” *Sensors* **19**(24), 5505 (2019).

14. M. Liu, X. Yin, E. Ulin-Avila, B. Geng, T. Zentgraf, L. Ju, F. Wang, and X. Zhang, “A graphene-based broadband optical modulator,” *Nature* **474**(7349), 64-67 (2011).

15. T. Kampfrath, L. Perfetti, F. Schapper, C. Frischkorn, and M. Wolf “Strongly coupled optical phonons in the ultrafast dynamics of the electronic energy and current relaxation in graphite,” *Phys. Rev. Lett.* **95**(18), 187403 (2005).

16. Z. Wang, J. Qiao, S. Zhao, S. Wang, C. He, X. Tao, and S. Wang, “Recent progress in terahertz modulation using photonic structures based on two-dimensional materials,” *InfoMat* **3**(10), 1110-1133 (2021).

17. M. Chen, P. Sheng, W. Sun, and J. Cai, “A symmetric terahertz graphene-based hybrid plasmonic waveguide,” *Opt. Commun.* **376**, 41-46 (2016).

18. W. Xu, Z. H. Zhu, K. Liu, J. F. Zhang, X. D. Yuan, Q. S. Lu, and S. Q. Qin, “Toward integrated electrically controllable directional coupling based on dielectric loaded graphene plasmonic waveguide,” *Opt. Lett.* **40**(7), 1603-1606 (2015).

Query 5. Fig. 2a: Add H and Lambda in the figure when presenting the geometry.

Q5. Author Action:

A5. Fig. 2(a) has been revised to clarify the definition of “ridge height” and “period length” of the proposed WBGs. (see Fig. 2(a) in the Author Action responding to your Query #6)

Query 6. Fig. 2b: It is hard to decipher the figures as there are too many curves, and there are not distinguishable (the squares and triangles look the same). I suggest the researchers consider splitting the individual figures in 2 (one on top and one on bottom). It is absolutely hard to understand in its current form. The same comment can be made for Fig. 2cd.

Q6. Author Response:

We agree with the reviewer that the original Fig. 2 is confusing as it includes some non-essential information about the WBG optical properties. In the revised Fig. 2 we retained only the key WBG spectral responses that are critical for the understanding of

the functioning of the ADMs. The non-essential information has been now moved to Supplementary Note 4 entitled “*Design of two-wire waveguide Bragg gratings*”. We hope that the new form of Fig. 2 is considerably easier to read.

Q6. Author Action:

A6. The revised Fig. 2 now shows the relative transmission coefficients (by field) of WBGs ($\Lambda=1.03$ mm, $H=0.21$ mm) having different numbers of periods for THz light propagating in the forward direction, and the relative reflection coefficients (by field) of the WBG (20 periods, $\Lambda=1.03$ mm, $H=0.21$ mm) with two possible THz light propagation directions, which are important for the design of ADMs studied in Chapter 2.3. In addition, the figure shows the relative transmission coefficients (by field) of WBGs as a function of the period length Λ , which verifies the spectral tunability of WBG and ADM.

Fig. 2 Two-wire Waveguide Bragg Grating. (a) Schematic of the WBG cross-section, its side view, and a photograph of the fabricated WBG with the top half removed. The grating featuring a periodic sequence of truncated cones (light gray) and a cylindrical wire (dark gray) is suspended in the air on dielectric supports and encapsulated within a plastic cage. The numerical and experimental relative transmission coefficients (by field) of the 2.5cm-long WBG sections (b) comprising a different number of periods ($\Lambda=1.03$ mm, $H=0.21$ mm), and (c) containing 20 periods ($H=0.21$ mm) as a function of the period length Λ . The THz light travels in the forward direction (see Fig. 2(a)). (d) The numerical and experimental relative reflection coefficients (by field) of the 2.5cm-long WBG sections comprising 20 periods ($\Lambda=1.03$ mm, $H=0.21$ mm) with two possible traveling directions of the THz light (see Supplementary Note 4 for more details).

B6. A new Fig. S5 that contains some of the information from the original Fig. 2 is placed into Supplementary Note 4 entitled “*Design of two-wire waveguide Bragg gratings*”. In particular, it shows the experimental transmission coefficients of a WBG (20 periods, $\Lambda=1.03$ mm, $H=0.21$ mm) for two possible THz light propagation directions, the numerical reflection coefficients of WBGs (20 periods, $H=0.21$ mm) featuring different period lengths Λ , and experimental raw data for the WBG scattering parameter S_{11} .

Fig. S5 Two-wire waveguide Bragg gratings. (a) Relative transmission coefficient (by field) and (b) Measured scattering parameter (S_{11}) of a 2.5cm-long WBG section comprising 20 periods ($\Lambda=1.03$ mm, $H=0.21$ mm) for THz light with two possible traveling directions. (c) Relative reflection coefficients (by field) and (d) Measured scattering parameter (S_{11}) of 2.5cm-long WBGs containing 20 periods ($H=0.21$ mm) as a function of the period length Λ . The THz light travels in the forward direction along WBG (see Fig. 2(a)).

Query 7. Line 161: ...notoriously difficult to eliminate. Needs a reference.

Q7. Author Response:

What makes detecting the actual spectral response of the WBGs and the ADM circuits a challenging task are the standing waves that are formed in the cavity of an optics-based free-space CW-THz spectroscopy setup, as well as inside of a photomixer silicon lens. Moreover, in the case of transmission measurements in the infrared spectral range one normally employs isolators between the device and a source to eliminate standing waves in the coupling section of a setup. Unfortunately, due to the lack of the THz isolators, resonances in the coupling section can also contribute to spurious oscillations in the transmission signal. Finally, due to the relatively low powers of an optics-based setup, the measurement data presented in the original paper was noisy, thus spurious features were difficult to eliminate even when dividing the measured spectra by the one of a properly chosen reference.

To address these challenges and to ameliorate our measurements we employed a network analyzer and solid-state THz sources with higher output powers, which belong to another research center in the Montreal area, to recharacterize the components and circuits of the current paper. For the sake of completeness, we retained the description of the optics-based CW-THz spectroscopy system and device characterization measurements by such a system, while moving them into Supplementary Note 9 entitled “Characterization of two-wire waveguide-based Bragg gratings and ADM circuits using free-space continuous-wave (CW) THz spectroscopy system”. At the same time, the BER measurements were still performed using an optics-based system as we could not find any solid-state system in the Montreal area that could perform such measurement at ~ 150 GHz. The newly obtained experimental results for the WBGs and grating-loaded side couplers show very similar spectral responses to the ones obtained using an optics-based system, however with dramatically reduced noise contribution, which can be seen in the revised Figs. 2 and 5.

Q7. Author Action:

The following references that study the influence of standing waves on measurement using this CW-THz spectroscopy system have been cited:

“Although such spectral oscillations can be somewhat mitigated via normalization with respect to a properly chosen reference, they are, nevertheless, notoriously difficult to eliminate [21,22].”

Query 8. Fig. 3a and 4a. The figure is confusing, the through beam should go from left to right.

Q8. Author Action:

A8. The schematics shown in Fig. 1, Fig. 3, and Fig. 4 have been modified. In the revised version, the In (Input) port of the ADM (directional coupler) circuit is on the left side to ensure that the THz beam goes from left to right.

Query 9. Lines 280-290: The difference between Balanced and Optimal designs is not clear, should be emphasized more, especially since it becomes important later in the text.

Q9. Author Action:

A9. In the third to fifth paragraphs of the revised Chapter 2.3, we emphasized the reason behind the difference in the optical performance of the two grating-loaded side couplers and then clarified the goals achieved by these two designs:

"It should be noted that the relative position of a 20.6mm-long grating inside the 35mm-long side coupler section (controlled by the L_d^{in} parameter) gives rise to different interference conditions between the two supermodes of a coupler, which we exploit to find optimal coupler designs. Particular attention is paid to the design and characterization of two grating-loaded side coupler sections that enable ADM circuits with either an optimized Drop action or a balanced Add/Drop action.

To optimize the Drop action we choose L_d^{in} that results in the highest power at the Drop port (Plane 1) for the carrier frequency at the center of the grating stopband. Defining L_g to be the light penetration distance into grating, the optimal position of the grating section L_d^{in} can be written as:

$$L_d^{in} + L_g \approx L_c/2 \quad (3)$$

Under this condition, the destructive interference of the two reflected supermodes at the In port (Plane 1) ensures the maximal power of the THz signal at the Drop port (Plane 1) [46-47]. Another way to optimize the ADM design is to reduce the difference between spectral performances of the Add and Drop actions at the expense of their absolute performance, thus resulting in a balanced ADM performance. That means that a similar output power will be registered at the Drop (Plane 1) and the Through (Plane 2) ports when the broadband THz signal is launched into the In (Plane 1) or the Add (Plane 2) ports correspondingly. In this optimization, one minimizes the difference between transmitted powers in the Through port under Add port launching and in the Drop port under the In port launching."

46. C. Riziotis, and M. N. Zervas, "Novel full-cycle-coupler-based optical add-drop multiplexer and performance characteristics at 40-Gb/s WDM networks," J. Light. Technol. **21**(8), 1828 (2003).

47. C. Riziotis, and M. N. Zervas, "Performance comparison of Bragg grating-based optical add-drop multiplexers in WDM transmission systems," IEE P-Circ Dev Syst **149**(3), 179-186 (2002).

Query 10. Line 301: While in this work, we use relatively lossy Y splitters comprising straight and curved branches. This is not a sentence.

Q10. Author Action:

In the revised version we eliminated this sentence altogether to avoid redundancy.

Query 11. Fig. 5: It is hard to follow where the inputs and outputs are. Small schematics showing In and Out ports for the figures should be added in all subpanels.

Q11. Author Action:

A11. To clarify the individual channel functionality in the ADM circuit, in Fig. 4(a) we have marked the appropriate input and output ports for the Drop, Add, and Through actions.

B11. We have modified Figs. 5(a-c) to add color-coded schematics showing the optical paths of the THz signals propagating through the grating-loaded side coupler under different actions. Additionally, we have moved the discussion of the insertion losses (see Fig. S11) into Supplementary Note 8 as this data is a simple derivative of transmission data presented in Fig. 5.

Fig. 5 Optimized grating-loaded side couplers. Power transmittances of the grating-loaded side couplers for (a) an ADM with the optimized Drop action ($L_d^{in}=1.5$ mm) and (b) an ADM with the balanced Add/Drop action ($L_d^{in}=6.9$ mm) under the In port launching (Plane 1) and the Add port launching (Plane 2) conditions. (c) Power transmittances of the two grating-loaded side couplers at the Through port (Plane 2) under the In port launching (Plane 1) condition. Inserts schematically show the signal optical paths corresponding to various ADM actions.

Query 12. Line 433: It is mentioned that better fabrication strategies can be used. Would making everything in metal rather than 3d printing be better?

Q12. Author Response:

This is indeed a very interesting question. From the fabrication point of view, partial metallization of 3D-printed plastic support is considerably easier than CNC machining of complex 3D circuitry from a metal slab. This is due to deeply sub-mm features of the waveguide support and grating structure, as well as the 3D complexity of a design. Alternatively, one might attempt fabrication of the all-metallic structures using additive manufacturing approaches such as selective sintering of metallic powders. To date, however, the resolution of the metal powder-based printers is in the 100-250 um range, which is much less performant than the 10-50 um resolution of the resin-based SLA printers.

Fig.S1 Electric field distributions of several lowest-loss plasmonic modes at 140 GHz supported by a hollow enclosure with metallic walls (a rectangular waveguide) containing two metallic wires (a) mode 1 (loss $1.6 m^{-1}$), (b) mode 2 (loss $2.2 m^{-1}$), (c) mode 3 (loss $2.4 m^{-1}$), (d) mode 4 (loss $2.6 m^{-1}$). Electric field distributions of several lowest-loss plasmonic modes at 140 GHz supported by an identical hollow enclosure with infinite plastic walls containing two metallic wires (e) mode 1 (loss $5 m^{-1}$), (f) mode 2 (loss $45 m^{-1}$), (g) mode 3 (loss $52 m^{-1}$), (h) mode 4 (loss $68 m^{-1}$). The loss is by field.

Additionally, all-metallic encapsulation of the two-wire waveguides can pose significant challenges due to the presence of a large number of higher-order, relatively low-loss modes that can cause multi-mode interference effects and bring about additional noise and artifacts to the operation of an ADM circuit. This is because a hollow metallic encapsulation cage is essentially a heavily multimode rectangular waveguide that supports a large number of relatively low-loss modes even when two metallic wires are placed inside. In Figs. S1(a-d) we present several of such plasmonic modes and their losses computed using the Impedance Boundary Conditions (see Methodology) for all the metallic walls. From this calculation, we see that for waveguides shorter than 1 m, all these modes will persist inside the waveguide and cause multimode interference. In comparison, by substituting metallic

walls with the infinite resin ones (see Figs. S1(e-h)), we observe that only a single mode with the highest field concentration in the gap between the two wires has a relatively low loss, while all the other modes have much higher losses. Thus, using a resin cage in place of a metallic one reduces the number of low-loss higher-order modes, thus suppressing the multi-mode interference effects, resulting in essentially single-mode operation of an encapsulated two-wire waveguide.

For these reasons, we believe that partial metallization of the 3D-printed resin-based structures is currently the best option for the fabrication of complex plasmonic THz circuits.

Q12. Author Action:

A12. We added Supplementary Note 2 entitled “*Alternative fabrication routes*” to address the issue of all-metal structures. It is based on the Author's response above.

B12. We also specified the further optimization strategy in the Discussion Chapter.

“Further optimization of the ADM circuits is possible by employing more efficient Y-splitter designs and resorting to better fabrication strategies, which include using higher-resolution 3D printers and more advanced metal deposition techniques for metallization. Additionally, coupling losses to detector and emitter can be further reduced via geometry optimization of the in-coupling and out-coupling ports via mode shaping. We estimate that a combined effect of such optimizations could reduce the insertion loss by 5-10 dB, while also improving and balancing the performance of the Add and Drop actions, thus rendering ADM devices highly suitable for the ultra-fast analog signal processing in THz communications.”

Query 13. The legends of all the figures are too long and hard to follow. They should be kept simple and straight to the point with minimal detail (especially since it is sometimes the same geometric details in all the subpanels).

Q13. Author Action:

A13. The legends of all the figures in the main text have been revised and shortened to convey only the essential information:

Fig. 1 Schematic of the four-port ADM circuit comprising a grating-loaded side coupler placed between two Y-splitters. L_d^{in} and L_d^{out} are the distances between the grating and the directional coupler end-facets (see an Insert in the red dotted box). The solid and hollow orange arrows indicate the paths of the dropped and added signals within the grating stopband. The green arrows indicate the path of a through signal in the grating passband. Insert in the blue dotted box presents a cross-section of the grating-loaded side coupler. Insert in the black dotted box shows a fabricated four-port ADM circuit with the top half removed.

Fig. 2 Two-wire Waveguide Bragg Grating. (a) Schematic of the WBG cross-section, its side view, and a photograph of the fabricated WBG with the top half removed. The grating featuring a periodic sequence of truncated cones (light gray) and a cylindrical wire (dark gray) is suspended in the air on dielectric supports and encapsulated within a plastic cage. The numerical and experimental relative transmission coefficients (by field) of the 2.5cm-long WBG sections (b) comprising a different number of periods ($A=1.03$ mm, $H=0.21$ mm), and (c) containing 20 periods ($H=0.21$ mm) as a function of the period length A . The THz light travels in the forward direction (see Fig. 2(a)). (d) The numerical and experimental relative reflection coefficients (by field) of the 2.5cm-long WBG sections comprising 20 periods ($A=1.03$ mm, $H=0.21$ mm) with two possible traveling directions of the THz light (see Supplementary Note 4 for more details).

Fig. 3 Two-wire directional coupler. (a) Schematic of the coupler cross-section featuring two touching two-wire waveguides that are suspended in the air on dielectric supports and encapsulated within a plastic cage. (b) A photo of the fabricated directional coupler circuit (top half removed) with a side coupler section placed between two Y-splitters. The black dashed lines show interfaces between distinct modular sections of the experimental circuit. Insert: the enlarged view of the Y junction where curved and straight waveguides meet. (c) The numerical effective refractive indices of the odd and even supermodes guided by the directional coupler, and the corresponding coupling length. Inserts: electric field distributions (vertical component) for the even and odd supermodes. (d) 3D modeling of the directional coupler circuits, and the corresponding electric field distributions when using the fundamental mode of Input port 1 at 140 GHz as the excitation condition. (I) Case of the half-cycle coupler that results in the maximal power transfer from Waveguide 1 (WG1) to Waveguide 2 (WG2). (II) Case of the full-cycle coupler that results in the maximal power transfer from Waveguide 1 to Waveguide 2 and then back to Waveguide 1. (e) The power transmittance ratio for the Output ports 1 and 2 at Plane 2 as a function of the side coupler section length $L_{coupler}$.

Fig. 4 Numerical study of the grating-loaded side couplers. (a) (I) 3D model of the ADM circuit used in simulations that comprises two Y-splitters and a 35mm-long grating-loaded side coupler (20 grating periods with $A=1.03$ mm, $H=0.21$ mm). The computed electric field distributions in the ADM circuits for various launching conditions: (II) In port at 140 GHz (grating stopband), (III) In port at 128 GHz (grating passband), and (IV) Add port at 140 GHz (grating stopband). Optimized Drop Action: reflectance and transmittance of various ports under (b) In port launching (c) Add port launching. Balanced Add/Drop Action:

reflectance and transmittance of various ports under (d) In port launching (e) Add port launching. Figs. 4(b-e) are computed using Eq. 1 and data from Fig. 4(a).

Fig. 5 Optimized grating-loaded side couplers. Power transmittances of the grating-loaded side couplers for (a) an ADM with the optimized Drop action ($L_d^{in}=1.5$ mm) and (b) an ADM with the balanced Add/Drop action ($L_d^{in}=6.9$ mm) under the In port launching (Plane 1) and the Add port launching (Plane 2) conditions. (c) Power transmittances of the two grating-loaded side couplers at the Through port (Plane 2) under the In port launching (Plane 1) condition. Inserts schematically show the signal optical paths corresponding to various ADM actions.

Fig. 6 Experimental study of the ADM circuits for THz communications. Eye patterns for the THz signals of different bit rates and different carrier frequencies propagating through (a) an ADM circuit with the optimized Drop action and (b) an ADM circuit with the balanced Add/Drop action. The measurements were performed at the following ports: (I-III) Drop port, (IV) Through port (Drop action: In port launch within grating stopband); (V-VII) Through port, (VIII) In port (Add action: Add port launch within grating stopband); (IX-XI) Through port, (XII) Drop port (Through action: In port launch within grating passband). The measurements are performed at various frequencies in the grating stopband (143 GHz, 148 GHz, 151 GHz) and the grating passband (128 GHz), which are also marked by black stars in Fig. 5. (c) Measured BER versus bit rate of the THz carrier wave propagating through two optimized ADM circuits.

Query 14. I am wondering if using two bare waveguides (with no gratings) would show similar performance in communications? In short, how are the authors convinced that what they are observing experimentally is really due to the effect of their gratings.

Q14. Author Response:

Compared to the ADM performance, the directional coupler circuits (without a grating) show very different optical performance. We have confirmed this experimentally and added Supplementary Note 10 entitled “Characterization of the directional coupler circuit using THz communication system” to address this issue.

Fig. S15 Experimental study of the directional coupler circuit shown in Fig. 3(b) for THz communications. Eye pattern of the THz signal that is launched into the Input port 1 and registered at the Output port 1 with the carrier frequency of (a) 128 GHz and (b) 143 GHz. Eye pattern of the THz signal which is launched into the Input port 1 and registered at the Input port 2 with the carrier frequency of (c) 128 GHz and (d) 143 GHz. (e) Eye diagram for the noise level of the THz communication system (no signal). (f) Measured BER versus bit rate for the THz signal propagating through the directional coupler circuit at 128 GHz and 143 GHz carrier frequencies.

Particularly, we find experimentally that regardless of the operation frequency (128 GHz or 143 GHz), a THz signal launched into the Input port 1 of a directional coupler (see Fig. S15) is forwarded mainly into the Output port 1 (a would-be Through port in the corresponding ADM circuit) with only a negligible portion arriving at the Input port 2 (a would-be Drop port in the corresponding ADM circuit). This is in stark contrast with the experimental behavior of ADM circuits shown in Fig. 5, where only a signal at 128 GHz (grating passband) reaches the Through port, while the signal at 143 GHz (grating stopband) is completely blocked from the Through port and redirected into the Drop port.

Q14. Author Action:

A14. Supplementary Note 10 entitled “Characterization of the directional coupler circuit using THz communication system” was added to address the issue brought by the reviewer. It includes Fig. S15 and the following text:

“To verify that the Add/Drop functionality of the ADM circuits presented in this work is due to Bragg gratings, we conducted a comparative study of a stand-alone directional coupler circuit (without a grating). Particularly, we find experimentally that

regardless of the operation frequency (128 GHz or 143 GHz), a THz signal launched into the Input port 1 of a directional coupler (see Fig. S15) is forwarded mainly into the Output port 1 (a would-be Through port in the corresponding ADM circuit) as seen in Figs. S15(a) and S15(b). At the same time, only a negligible portion of power is redirected into the Input port 2 (a would-be Drop port in the corresponding ADM circuit) as seen from Figs. S15(c) and S15(d). A small signal in the Input port 2 of a directional coupler circuit is only slightly above the THz communication system noise level (Fig. S15(e)) and is due to crosstalk between two branches of the Y-splitter. This is in stark contrast with the experimental behavior of the ADM circuits shown in Fig. 5, where only a signal at 128 GHz (grating passband) reaches the Through port, while the signal at 143 GHz (grating stopband) is completely blocked from the Through port and redirected into the Drop port.

Moreover, due to the broadband operation of the directional coupler (see Fig. 3(c)), similar BER performance characteristics (see Fig. S15(f)) are observed at the Output port 1 of the directional coupler for the two frequencies. In fact, the BER curves presented in Fig. S15(f) are very similar to those of a Through action for the complete ADM circuits presented in Fig. 6(c). This is easy to rationalize as the Through action of an ADM circuit is designed for operation outside of the grating stopband, thus being only affected by the directional coupler performance.”

B14. A note was to the first paragraph of Chapter 2.4.

“Also, to confirm the key role of the WBGs for the functioning of ADM circuits, a complimentary experimental study of the directional coupler circuit without a grating is presented in Supplementary Note 10.”

Reference

1. G. Biagi, J. Fiutowski, I. P. Radko, H. G. Rubahn, K. Pedersen, and S. I. Bozhevolnyi, "Compact wavelength add-drop multiplexers using Bragg gratings in coupled dielectric-loaded plasmonic waveguides," *Opt. Lett.* **40**(10), 2429-2432 (2015).
2. K. Nallappan, Y. Cao, G. Xu, H. Guerboukha, C. Nerguizian, and M. Skorobogatiy, "Dispersion-limited versus power-limited terahertz communication links using solid core subwavelength dielectric fibers," *Photonics Res.* **8**(11), 1757-1775, 2020.
3. S. Carpenter, D. Nopchinda, M. Abbasi, Z. S. He, M. Bao, T. Eriksson, and H. Zirath, "A D-band 48-Gbit/s 64-QAM/QPSK direct-conversion I/Q transceiver chipset," *IEEE Trans. Microw. Theory Techn.* **64**(4), 1285-1296 (2016).
4. M. Fujishima, M. Motoyoshi, K. Katayama, K. Takano, N. Ono, and R. Fujimoto, "98 mW 10 Gbps wireless transceiver chipset with D-band CMOS circuits," *IEEE J. Solid-State Circuits* **48**(10), 2273-2284 (2013).
5. Y. Cao, K. Nallappan, H. Guerboukha, G. Xu, and M. Skorobogatiy, "Additive manufacturing of highly reconfigurable plasmonic circuits for terahertz communications," *Optica* **7**(9), 1112-1125 (2020).
6. I. Kallfass, I. Dan, S. Rey, P. Harati, J. Antes, A. Tessmann, S. Wagner, M. Kuri, R. Weber, H. Massler, A. Leuther, T. Merkle, and T. Kurner, "Towards MMIC-based 300GHz indoor wireless communication systems," *IEICE Trans Inf Syst* **98**(12), 1081-1090 (2015).
7. W. Deal, X. B. Mei, K. M. Leong, V. Radisic, S. Sarkozy, and R. Lai, "THz monolithic integrated circuits using InP high electron mobility transistors," *IEEE Trans. Terahertz Sci. Technol.* **1**(1), 25-32 (2011).
8. H. J. Song, J. Y. Kim, K. Ajito, N. Kukutsu, and M. Yaita, "50-Gb/s direct conversion QPSK modulator and demodulator MMICs for terahertz communications at 300 GHz," *IEEE Trans. Microw. Theory Techn.* **62**(3), 600-609 (2014).
9. Y. Cao, K. Nallappan, H. Guerboukha, T. Gervais, and M. Skorobogatiy, "Additive manufacturing of resonant fluidic sensors based on photonic bandgap waveguides for terahertz applications," *Opt. Express* **27**(20), 27663-27681 (2019).
10. M. Poulin, S. Giannopoulos, and M. Skorobogatiy, "Surface Wave Enhanced Sensing in the Terahertz Spectral Range: Modalities, Materials, and Perspectives," *Sensors* **19**(24), 5505 (2019).
11. M. Liu, X. Yin, E. Ulin-Avila, B. Geng, T. Zentgraf, L. Ju, F. Wang, and X. Zhang, "A graphene-based broadband optical modulator," *Nature* **474**(7349), 64-67 (2011).
12. T. Kampfrath, L. Perfetti, F. Schapper, C. Frischkorn, and M. Wolf, "Strongly coupled optical phonons in the ultrafast dynamics of the electronic energy and current relaxation in graphite," *Phys. Rev. Lett.* **95**(18), 187403 (2005).
13. Z. Wang, J. Qiao, S. Zhao, S. Wang, C. He, X. Tao, and S. Wang, "Recent progress in terahertz modulation using photonic structures based on two-dimensional materials," *InfoMat* **3**(10), 1110-1133 (2021).
14. M. Chen, P. Sheng, W. Sun, and J. Cai, "A symmetric terahertz graphene-based hybrid plasmonic waveguide," *Opt. Commun.* **376**, 41-46 (2016).
15. W. Xu, Z. H. Zhu, K. Liu, J. F. Zhang, X. D. Yuan, Q. S. Lu, and S. Q. Qin, "Toward integrated electrically controllable directional coupling based on dielectric loaded graphene plasmonic waveguide," *Opt. Lett.* **40**(7), 1603-1606 (2015).

REVIEWER COMMENTS

Reviewer #1 (Remarks to the Author):

In order to deserve publication in a high-impact journal, an electromagnetic device must either involve some scientifically interesting and novel physical principles or exhibit practical advantages. Following careful review of the revised manuscript, it is clear that neither is true in this case, for reasons given below. Thus, I am regretfully unable to recommend this manuscript for publication in Nature Communications.

R1.1 The authors have indeed improved the agreement between experiment and simulation, but the fact of the matter is that the achieved performance of this device is less than impressive. Fig. 5 shows low efficiency of <40% in experiment, as well as poor channel quality (i.e. jagged frequency response, instead of a contiguous band of frequencies that exhibits minimal ripple or variation), and the efficiency of the “through” path is <20%. This performance is below academic standard.

R1.2 The authors have confirmed that their ADM uses essentially the same operation principle that was demonstrated half a decade prior in [G. Biagi et. al. Opt. Lett. 40, 2429-2432 (2015)]. Furthermore, they have previously disseminated their 3D-printed plasmonic mm-wave waveguide structure [Y. Cao et. al., Optica 7(9), 1112-1125 (2020)]. The combination of these two concepts does have some small novelty, but this is insufficient for publication in Nature Communications.

R1.3 The authors have confirmed that, in the communications experiment, the reduced performance of the “add” and “drop” signal paths is due to the intrinsic limitations of the device itself; these channels have markedly reduced bandwidth and overall efficiency in comparison to the “through” path. This casts further doubt upon the value of the device as a multiplexer.

R1.5 The large size of the presented device is a significant disadvantage, and the authors do not make a very strong argument for the aspects in which their two-wire solution outperforms existing integrated solutions.

Reviewer #2 (Remarks to the Author):

I want to thank the authors for addressing my specific comments. I appreciate the new improved manuscript which I found to be easier to read. I have some additional minor comments that must be addressed, the main one being to change the units in the figure to dB scale. Please see below. To answer the authors' comments on my specific comments:

1. OK

2. OK. I am satisfied with the new additions to the introduction.

3. OK

4. I thank the authors for the new experiments regarding the thermal tunability. However, I regret to say that the experimental results are not convincing. There is no clear change in the minimum position as the temperature increases. I believe the main problem here was the resin which did not have a large enough thermal expansion coefficient. I believe other resins can be chosen to meet that goal. I think the authors should 1) recognize the limit of the thermal tunability of their prototype (in the paper and/or supplementary), and 2) state that alternative resin materials could be used to achieve this goal.

5. OK

6. I appreciate the new Fig. 2. It conveys the important information in a more concise manner. However, I believe the units should be in dB scale (power) in all figures. The authors use dB values at many places when discussing the text. For cohesiveness, the same dB scale should be used in all the figures.

7. OK

8. OK

9. OK. The difference between drop and balanced add/drop design is better explained.

10. OK

11. OK. The added schematics explain better Fig. 4.

12. OK. The authors' argument about multimode propagation is convincing and supported by numerical simulations.

13. OK.

14. OK. I thank the authors for the new experiments involving a straight waveguide. They are convincing. Small note: in the new text in the paper, "complimentary" should be "complementary".

Additional comment: I have been aware of a recently published article in Nature Communications with a topic similar to the work by Cao et al.:

Dong, J., Tomasino, A., Balistreri, G. et al. Versatile metal-wire waveguides for broadband terahertz signal processing and multiplexing. Nat Commun 13, 741 (2022). <https://doi.org/10.1038/s41467-022-27993-7>

This was published on Feb. 8, 2022 (which I assume had a review schedule similar to this work by Cao et al.). I note that in the work by Dong et al., the authors study a wire-based THz waveguide for signal processing and multiplexing. While the topic is similar, there are still, I believe, substantial differences between the 2 works. 1) Dong et al. did not perform any communication experiments and only characterized their system with a time domain spectrometer, 2) Dong et al. proposed a waveguide geometry for polarization division multiplexing, while Cao et al. proposed a waveguide for frequency division multiplexing. Following these observations, I believe there is still room for publication for the manuscript by Cao et al. especially since they were submitted around the same time. In my opinion however, the authors from Cao et al. should at least recognize this paper in their manuscript by properly citing it.

First, we would like to thank the reviewers and editors for their time and thoughtful comments. Our detailed reply to address the reviewers' concerns is as follows. Additionally, we also revised the explanation and figures in both the main text and supplementary notes to illustrate our work clearly. For convenience, we indicate in **bold** the original questions by the reviewer, in regular font our response to the reviewer, and in *italic* our corrections to the manuscript.

Reviewer 1:

R1 The authors have indeed improved the agreement between experiment and simulation, but the fact of the matter is that the achieved performance of this device is less than impressive. Fig. 5 shows low efficiency of <40% in experiment, as well as poor channel quality (i.e., jagged frequency response, instead of a contiguous band of frequencies that exhibits minimal ripple or variation), and the efficiency of the “through” path is <20%. This performance is below academic standard.

Author Response:

We admit the imperfect efficiencies for channel Drop and Through actions, as well as jagged frequency response for channel Drop action in certain instances. However, the statement that the performance of the proposed THz add-drop multiplexer (ADM) is below academic standard has to be nuanced.

Taking the optimized Drop action as an example (see Fig. 5(a)), its insertion loss (efficiency of ~40%) mainly stems from the transmission loss of the THz signal propagating along straight two-wire waveguides (~1.5 dB for ~3cm-length optical path [1]) and the incomplete reflection of Bragg gratings (~2.5 dB insertion loss, see Fig. 2) inside the grating-loaded side coupler section. Compared with the well-developed components for mid-IR (ex. single-mode silica fiber featuring transmission loss of 0.25 dB/km and fiber Bragg grating containing thousands of periods with a reflectance above 99% [2]), the two-wire THz plasmonic waveguides and Bragg gratings suffer much higher insertion losses. Actually, due to the lack of trusted material platforms, universal standards for THz communication components, as well as the relative abundance of potential, while untested, design and fabrications routes, it is not surprising that most of the reported THz waveguide components are far less efficient than the ones with similar structures operating in the mid-IR and microwave. While in the field of THz optical devices, compared with the ones built on other kinds of THz waveguide platforms (ex. rectangular waveguide and solid-core fiber in dielectric featuring low material absorption [3], both with transmission loss ~0.2 cm⁻¹ for THz signal with operating frequency below 0.5 THz), the two-wire waveguide features relatively low transmission loss (~0.12 cm⁻¹) and negligible dispersion, and the proposed two-wire waveguide Bragg grating enables relatively strong reflection. Additionally, the efficiency for Add and Drop actions of the proposed two-wire ADM is on the same level as the reported THz multiplexers employing other working mechanisms for frequency-division multiplexing (FDM) (see Table R1). Therefore, we believe that in the THz regime, the proposed two-wire ADM enables fairly efficient Add and Drop actions.

In terms of the Through action, its insertion loss of ~6.5 dB is mainly attributed to the mismatch between the modal fields of straight waveguide and gratings, as well as the scattering loss due to the directional coupling between two waveguides inside the grating-loaded side coupler section. Such inefficiency is an intrinsic defect for the ADM employing a similar configuration (ex. the experimental efficiency is below 15% for the mid-IR ADM built on a planar dielectric waveguide [4]). Despite this high insertion loss, channel Through action was still demonstrated in this work, thus confirming the Through function of this ADM for THz FDM communications. Additionally, we found that some developed THz multiplexers do not support Through action, but this cannot deny their impact and potential applications in THz

communications, which is supported by the high citation of corresponding reports (see Table R1). In terms of the stability of the spectral response, certain ADMs show somewhat jagged spectral response for Add and Drop actions indeed. However, the statement of poor channel quality for the proposed THz ADMs has to be nuanced. As the destructive interference of the two reflected supermodes at In port (Plane 1), maximal power of the THz signal can be observed at Drop port (Plane 1) of the ADM enabling optimized Drop action, thus resulting in a smooth 12GHz-bandwidth peak-shaped power transmittance spectra within the whole grating stopband (see Fig. 5(a)). Although one can observe ripples on the transmittance spectra for Add and Drop actions of ADMs with other structures, it should be noted that these designs are not aimed at achieving smooth spectral response, but proving the designability of Add and Drop actions as their transmittance spectra within the grating stopband differing when the relative position of the grating section changes (see Figs. S8 and S9). To confirm this fact, another design, i.e., ADM with balanced Add and Drop actions, was demonstrated in this work (see Fig. 5(b)). The THz carriers falling in the spectral range of 136-140 GHz and 149-153 GHz can be dropped and added efficiently, while the one with the frequency of 143 GHz (in the middle of the grating stopband) can not be found at Drop and Through ports. Therefore, by adjusting the relative position of the grating section on directional couplers, the channel adding and dropping of the proposed two-wire ADMs can be contiguously efficient within the whole grating stopband for broadband THz signal, while can also be selectively efficient at certain spectral position within the grating stopband for narrowband channels. The channel quality of the developed ADM is not poor but tunable.

Finally, a detailed comparison of reported THz multiplexers for FDM is presented in Table R1. This is not intended to show that a particular ADM outperforms others on specific metrics, but to illustrate the current progress in the related field and reveal that each multiplexer has unique working mechanism, advantages, and limits. Due to the lack of universal platforms for THz beam steering and exceedingly performant device designs to split THz signals in spectral and spatial domains, we believe that the uniform and explicit academic standard to judge whether the relevant work can be published is not clear yet.

Reference

1. Y. Cao, K. Nallappan, H. Guerboukha, G. Xu, and M. Skorobogatiy, "Additive manufacturing of highly reconfigurable plasmonic circuits for terahertz communications," *Optica* 7(9), 1112-1125 (2020).
2. D. Arora, J. Prakash, H. Singh, and A. Wason, "Reflectivity and Bragg's wavelength in FBG," *International Journal of Engineering (IJE)* 5(5), (2011).
3. J. Anthony, R. Leonhardt, A. Argyros, and M. C. Large, "Characterization of a microstructured Zeonex terahertz fiber," *JOSA B* 28(5), 1013-1018 (2011).
4. G. Biagi, J. Fiutowski, I. P. Radko, H. G. Rubahn, K. Pedersen, and S. I. Bozhevolnyi, "Compact wavelength add-drop multiplexers using Bragg gratings in coupled dielectric-loaded plasmonic waveguides," *Optics Letter* 40(10), 2429-2432 (2015).

Author Action:

A1. We indicated the development status of THz ADMs, as well as the reasons for their inefficiency when compared with the infrared and microwave components at the end of the second paragraph in Introduction chapter.

"However, due to the lack of trusted material platforms, universal standards for THz communication components, as well as the relative abundance of potential, while untested, design and fabrications routes, the ADMs capable of operation at THz frequencies are still in the early development. At the same

time, most of the reported THz ADMs suffer from low efficiency when compared with the infrared and microwave components of similar configurations.”

A2. We indicated that the channel capacity for Add/Drop actions is not poor but tunable by analyzing the two-wire ADMs’ transmittances.

In the seventh paragraph of Chapter 2.3:

“Nevertheless, the relatively smooth 12GHz-bandwidth peak-shaped power transmittance spectrum confirms the efficient channel Drop action of this ADM for THz carriers that fall within the whole grating stopband.”

In the second paragraph of Supplementary Note 7:

“It is found that transmittance of Drop (Through) port under In (Add) port launching condition varies with the L_d^n parameter, thus confirming the designability of channel Drop (Add) response of ADM circuit. With proper designs, the channel adding and dropping of the proposed two-wire ADMs can be contiguously efficient within the whole grating stopband for broadband THz signal, while can also be selectively efficient at certain spectral position within the grating stopband for narrowband channels.”

A3. We added the following table in Supplementary Note 1 to briefly review the reported THz multiplexers for FDM communications.

Table R1 Reported THz multiplexers for FDM communications

Design	Dimension	Working principle	Function of single element	Performance (Insertion loss & bandwidth)	Unique advantages	Main limits	Reference
 PPWG	3D On the level of cm ³	Leaky-wave radiation through slot on a plate	Add/Drop	Above 12 dB with tunable bandwidth for Drop/Add action	 ✓ Broadband operation frequency ✓ Tunable bandwidth 	 □ Difficult in optical alignment □ Compromise between bandwidth and crosstalk 	Nature Photonics , 2015, 9(11) [8] Nature Communications , 2017, 8(1) [9] Citation: 206
 PPWG & Electrically actuated components	3D On the level of cm ³ For 100 GHz	Coupling between waveguides with the assistance of resonance in cavity	Drop & Through	2.5 dB with a bandwidth of ~10 GHz for Drop action & 2 to 6 dB within a broad band for Through action	 ✓ Relatively high efficiency ✓ Switchable output port for dynamic band allocation 	 □ Complicated structure due to the involvement of active element □ Jagged spectral response for Through action 	Nature Communications , 2018, 9(1) [12] Citation: 29
 Photonic crystal on silicon wafer	2D Less than 1mm ² For key elements at 300 GHz	Directional coupling between adjacent waveguides & Photonic bandgap effect	Drop & Through	6.2 dB with a bandwidth of 6.5 GHz for Drop action & ~6 dB with a bandwidth of 25 GHz for Through action	 ✓ Small size ✓ Enable tandem connection on a wafer 	 □ High group velocity dispersion □ Narrowband with restricted spectral range for both Drop and Through actions 	Optics Express , 2016, 24(7) [10] Advanced Optical Materials , 2018, 6(16) [11] Citation: 107
 Unclad planar silicon waveguide	2D 4 cm ² For 300 GHz	Leaky-wave radiation & Constructive interference	Drop	~3 dB with a bandwidth of ~15 GHz for Drop action	 ✓ Support multiple channels ✓ Low insertion loss 	 □ Complicated design □ Crosstalk between channels 	Optica , 2021, 8(5) [13] Citation: 7
 Two-wire waveguide	3D 0.33 cm ² ×3.5 cm For key elements at 100 GHz	Bragg reflection & Directional coupling	Drop & Add & Through	>4 dB with a bandwidth of <12 GHz for Drop & Add action & ~6.5 dB within broadband for Through action	 ✓ Modular element for reconfigurable circuit ✓ Easy to align with other devices and tunable action 	 □ Large size □ Require additional component for beam steering 	This work

8. N. J. Karl, R. W. McKinney, Y. Monnai, R. Mendis, and D. M. Mittleman, "Frequency-division multiplexing in the terahertz range using a leaky-wave antenna," *Nat. Photonics* 9(11), 717-720 (2015).
9. J. Ma, N. J. Karl, S. Bretin, G. Ducournau, and D. M. Mittleman, "Frequency-division multiplexer and demultiplexer for terahertz wireless links," *Nat. Commun.* 8(1), 1-8 (2017).
10. M. Yata, M. Fujita, and T. Nagatsuma, "Photonic-crystal diplexers for terahertz-wave applications," *Opt. Express* 24(7), 7835-7849 (2016).
11. W. Withayachumnankul, M. Fujita, and T. Nagatsuma, "Integrated silicon photonic crystals toward terahertz communications," *Adv. Opt. Mater.* 6(16), 1800401 (2018).
12. K. S. Reichel, N. Lozada-Smith, I. D. Joshipura, J. Ma, R. Shrestha, R. Mendis, M. D. Dickey, and D. M. Mittleman, "Electrically reconfigurable terahertz signal processing devices using liquid metal components," *Nat. Commun.* 9(1), 1-6 (2018).
13. D. Headland, W. Withayachumnankul, M. Fujita, and T. Nagatsuma, "Gratingless integrated tunneling multiplexer for terahertz waves," *Optica* 8(5), 621-629 (2021).

R2 The authors have confirmed that their ADM uses essentially the same operation principle that was demonstrated half a decade prior in [G. Biagi et. al. *Opt. Lett.* 40, 2429-2432 (2015)]. Furthermore, they have previously disseminated their 3D-printed plasmonic mm-wave waveguide structure [Y. Cao et. al., *Optica* 7(9), 1112-1125 (2020)]. The combination of these two concepts does have some small novelty, but this is insufficient for publication in *Nature Communications*.

Author Response:

Actually, to the best of our knowledge, the operation principle of the proposed two-wire THz plasmonic ADM circuits, i.e., imposing Bragg gratings on directional coupler section, was firstly demonstrated in 1996 on the platform of silica fiber [1]. This configuration has attracted the attention of researchers with interests in mid-IR communications over the past two decades [2-5] due to its inherent advantages of simple design, enabling tandem connection, as well as a relatively high tolerance for errors in the component structures. Meanwhile, the novel 3D printed two-wire plasmonic waveguide owns unique advantages over most reported THz waveguides (ex. low transmission loss, negligible dispersion, as well as modular and robust structure). In this work, a THz ADM is developed by judiciously combining the well-established operation principle and the novel waveguide platform, opening new perspectives for THz network design. Its novelties and uniqueness of novel plasmonic structure designs, robust device fabrication/prototyping techniques, modular subcomponent integration approach, intrinsic circuit tunability, and cost-effectiveness for THz band have been detailed in Chapter 1 and Supplementary Note 1. Herein, we intend to illustrate the development of most reported THz multiplexers and analyze the potential impact of the two-wire plasmonic circuit on THz communications to complement the novelties of our work.

As THz frequencies find themselves between infrared and microwave bands in the electromagnetic spectrum, most reported THz multiplexers benefit from either the mature optical or electronic solutions. As concluded in Table R1, some reported THz multiplexers built on PPWG and planar silicon wafers employ the operation principle of leaky-wave radiation, which are the combination of ordinary THz waveguide and the key concept of the well-developed leaky-wave antennas for microwave [6]. Based on the same platforms, other THz multiplexers employing directional coupling between two waveguides with/without the involvement of resonators draw on the configuration of well-developed optical components for mid-IR [7,8]. All of these works have been published in highly reputed journals (ex. *Nature Photonics*, *Nature Communications*, *Optica*, etc.) over the past few years. It should be noted that we are not trying to deny the novelties of these works or the strict requirement of these journals for the novelty but to reveal the general laws for the development of THz devices. Additionally, another work on two-wire waveguide component for THz polarization-division multiplexing, which was submitted to *Nature Communications* at almost the same time as our work (June 2021), was just published in February 2022 [9]. Based on the conventional stand-alone two-wire plasmonic waveguide, a four-wire component consisting of four wires arranged in a square geometry was developed to support polarization-division multiplexing of THz waves with two orthogonal polarization directions. Bragg gratings enabling frequency-dependent signal filtering were also developed. However, it should be noted that there are substantial differences between their work and ours as they explore distinctive routes to realize multi-channel THz communications (PDM vs. FDM) by taking advantage of different properties of the two-wire THz plasmonic waveguide platforms (linearly polarized mode vs. broadband operation and flexible structure). In addition, our work moves one step closer to achieving this goal by using THz carrier waves to experimentally demonstrate THz FDM communications and proposing a novel scalable platform that

can incorporate plasmonic devices of any size and complexity. Therefore, by comparing our work with the above-mentioned published reports, we believe it does not fall below publication standards in terms of novelty.

Furthermore, we believe that the impact of this work goes beyond combining a well-developed concept and a new THz plasmonic waveguide platform for a specific function of THz communications. It introduces a novel strategy to build customizable THz communication systems where the widely employed rectangular waveguide devices and the emerging plasmonic waveguide platform are integrated, and the THz signal propagates in a highly controlled environment. Due to the flexible structure and easily accessible modal fields of the proposed two-wire waveguide, modular THz optical components with advanced signal processing functionalities for THz communications can be developed, thus eventually forming a robust all-waveguide THz reconfigurable circuit where individual elements can be easily added, replaced, or subtracted. In principle, this circuit can be seamlessly integrated into the upcoming THz communication networks by completely forgoing free-space optics and complicated optical alignment. Considering the unique fabless fabrication route of this integrated optical circuit, we believe this work opens new perspectives for the development of cost-effective small-scale terahertz communication systems for mass production.

Reference

1. I. Baumann, J. Seifert, W. Nowak, and M. Sauer, "Compact all-fiber add-drop-multiplexer using fiber Bragg gratings," *IEEE Photonics Technology Letters* 8(10), 1331-1333 (1996).
2. C. Riziotis, and M. N. Zervas, "Novel full-cycle-coupler-based optical add-drop multiplexer and performance characteristics at 40-Gb/s WDM networks," *Journal of Lightwave Technology* 21(8), 1828 (2003).
3. C. Riziotis, and M. N. Zervas, "Performance comparison of Bragg grating-based optical add-drop multiplexers in WDM transmission systems," *Iee Proceedings-Circuits Devices and Systems* 149(3), 179-186 (2002).
4. D. Mechin, P. Grosso, and D. Bose, "Add-drop multiplexer with UV-written Bragg gratings and directional coupler in SiO₂-Si integrated waveguides," *Journal of Lightwave Technology* 19(9), 1282-1286(2001).
5. D. Mu, H. Qiu, J. Jiang, X. Wang, Z. Fu, Y. Wang, X. Jiang, H. Yu, and J. Yang, "A four-channel DWDM tunable add/drop demultiplexer based on silicon waveguide Bragg gratings," *IEEE Photonics Journal* 11(1), 1-8 (2019).
6. L. Goldstone, and A. A. Oliner, "Leaky-wave antennas I: Rectangular waveguides," *IRE Transactions on Antennas and propagation* 7(4), 307-319 (1959).
7. J. Zimmermann, M. Kamp, A. Forchel, and R. März, "Photonic crystal waveguide directional couplers as wavelength selective optical filters," *Optics Communications* 230(4-6), 387-392 (2004).
8. S. Fan, P. R. Villeneuve, J. D. Joannopoulos, and H. A. Haus, "Channel drop tunneling through localized states," *Physical Review Letters* 80(5), 960 (1998).
9. J. Dong, A. Tomasino, G. Balistreri, P. You, A. Vorobiov, É. Charette, B. L. Drogoff, M. Chaker, A. Yurtsever, S. Stivala, M. A. Vincenti, C. D. Angelis, D. Kip, J. Azaña, and R. Morandotti, "Versatile metal-wire waveguides for broadband terahertz signal processing and multiplexing," *Nature communications* 13(1), 1-8 (2022).

Author Action:

We have supplemented the potential impact of the proposed two-wire THz plasmonic circuits on THz communications in the sixth paragraph of Introduction Chapter.

"On the component integration level, the methods developed in this work show a pathway to the highly reconfigurable, modular construction of complex terahertz circuits, where individual components can be easily added, replaced, or subtracted to modify the device functionality. In addition to the FDM demonstrated in this work, the developed waveguide circuits can be also used in a four-wire

configuration to realize THz Polarization-Division Multiplexing [33], thus paving a way for supporting several modulation modalities in a single device. Furthermore, two-wire waveguides can be efficiently butt-coupled to rectangular waveguides, which are currently prevalent in microwave and mm-wave communication devices [29]. This allows two-wire waveguide-based optical components developed within this work to be, in principle, seamlessly integrated into the upcoming THz communication networks by completely forgoing free-space optics and complicated optical alignment.”

29. Y. Cao, K. Nallappan, H. Guerboukha, G. Xu, and M. Skorobogatiy, “Additive manufacturing of highly reconfigurable plasmonic circuits for terahertz communications,” *Optica* 7(9), 1112-1125 (2020).

33. J. Dong, A. Tomasino, G. Balistreri, P. You, A. Vorobiov, É. Charette, B. L. Drogoff, M. Chaker, A. Yurtsever, S. Stivala, M. A. Vincenti, C. D. Angelis, D. Kip, J. Azaña, and R. Morandotti, “Versatile metal-wire waveguides for broadband terahertz signal processing and multiplexing,” *Nat. Commun.* 13(1), 1-8 (2022).

R3 The authors have confirmed that, in the communications experiment, the reduced performance of the “add” and “drop” signal paths is due to the intrinsic limitations of the device itself; these channels have markedly reduced bandwidth and overall efficiency in comparison to the “through” path. This casts further doubt upon the value of the device as a multiplexer.

Author Response:

We admit that there are differences in the overall efficiency and bandwidth between channel Add/Drop and Through actions of the proposed full ADM circuit. However, the statement that it casts doubt upon the value of this device as a THz multiplexer has to be nuanced.

The efficiency difference between Channel Add/Drop and Through actions of the full two-wire THz plasmonic ADM circuit (>3 dB insertion loss difference) is larger than that of some conventional mid-IR ADMs (ex. the one employing Mach-Zehnder-based fiber gratings whose difference in the insertion loss for Drop and Through action is on the order of tenths of dB [1]). This fact is mainly due to the difference in the transmission loss of the employed waveguide platform (see more detailed information in Author Response to R1). Compared with the well-developed mid-IR waveguide platform (ex. silica-based planar lightwave circuits and fiber arrays), the moderate transmission loss of two-wire THz plasmonic waveguide (~0.4 dB/cm and ~1.25 dB/cm for straight and bent waveguides, respectively) gives rise to more pronounced insertion loss differences for signals propagating through optical paths of slightly different lengths, thus deteriorating the balance in the efficiency for channel Add/Drop and Through actions in our case. Nevertheless, the resultant imbalance remains manageable; both channel Add/Drop and Through functionalities have been confirmed using the proposed two-wire ADM in experiments (see Fig. 6).

In principle, one can simply use widely available electronic-based THz emitters with a few dBm output to mitigate the impact of this imbalance and demonstrate clear-channel communications. In addition, as clarified in Chapter 2.4, the imbalanced overall efficiency for Add/Drop and Through actions is attributed to the insertion loss difference of both the grating-loaded side coupler section (>4 dB and ~7 dB for channel Add/Drop and Through, respectively) and two Y-splitters (~10 dB and ~3.6 dB for channel Add/Drop and Through, respectively) inside the full ADM circuit. It is worth noting that this ADM configuration was designed for the convenience of characterizing the grating-loaded side coupler by THz spectroscopy systems, not aiming at building the trade-off between the efficiency for channel Add/Drop and Through actions. Actually, when integrating its dispensable element (i.e., grating-loaded side coupler section) in THz plasmonic circuits, the cumbersome branches of the two Y-splitters can be replaced by

other more efficient and compact waveguide components. With proper subcomponent design, the desired balance in efficiency (insertion loss) for channel Add/Drop and Through actions can be achieved. Furthermore, it is not surprising that channel Add/Drop action features a smaller bandwidth than the Through action of our proposed ADM circuit, which is also a general rule for most mid-IR ADMs. In dense WDM telecommunication networks, the narrow bandwidth and small spacing of dropped and added signals require the ADM to selectively process the carrier wave of a specific frequency, thus reducing the crosstalk between channels. Meanwhile, in order to extract/add multiple signals of different carrier frequencies to/from distinct ports of THz networks containing tandem ADMs, each ADM block needs to enable broad bandwidth for channel Through action. In most cases, mid-IR ADMs employ narrowband filtering elements (ex. Mach-Zehnder interferometers [1], ring resonators [2], and phase-shifted gratings [3]) and waveguide platforms supporting broadband operation to meet these requirements. A reported THz ADM enabling channel Through action also adopts this strategy [4]. Therefore, for the proposed two-wire THz ADM circuit, the imbalances in the overall efficiency and bandwidth for channel Add/Drop and Through actions would not diminish its value as a multiplexer for THz multi-channel communications.

Reference

1. T. Mizuochi, T. Kitayama, K. Shimizu, and K. Ito, "Interferometric crosstalk-free optical add/drop multiplexer using Mach-Zehnder-based fiber gratings," *Journal of Lightwave Technology* 16(2), 265-276 (1998).
2. Z. Qiang, W. Zhou, and R. A. Soref, "Optical add-drop filters based on photonic crystal ring resonators," *Optics Express* 15(4), 1823-1831 (2007).
3. H. A. Haus, and Y. Lai, "Narrow-band optical channel-dropping filter," *Journal of Lightwave Technology* 10(1), 57-62 (1992).
4. K. S. Reichel, N. Lozada-Smith, I. D. Joshupura, J. Ma, R. Shrestha, R. Mendis, M. D. Dickey, and D. M. Mittleman, "Electrically reconfigurable terahertz signal processing devices using liquid metal components," *Nature Communications* 9(1), 1-6 (2018).

Author Action:

The above-mentioned information has been concluded at the end of Supplementary Note 8:

"It is worth noting that such differences in the overall efficiency and bandwidth of channel Add/Drop and Through functions of the full THz ADM circuit can not diminish its value as a THz multiplexer for multi-channel communications. Indeed, the proposed full ADM configuration was designed for the convenience of characterizing its crucial element (i.e., grating-loaded side coupler section) by THz spectroscopy systems, not aiming at building the trade-off between channel Add/Drop and Through actions. Therefore, it is not surprising to observe such imbalances in our case, which is also common for mid-IR ADMs in telecommunication networks. For the proposed two-wire ADM, one can simply adjust the structure of its replaceable beam steering element (Y-splitters) and superimpose other grating structures (ex. uniform gratings with high index contrast (see Fig. S2(b)) on its indispensable directional coupler section to reduce the difference in the overall efficiency and bandwidth, respectively."

R5 The large size of the presented device is a significant disadvantage, and the authors do not make a very strong argument for the aspects in which their two-wire solution outperforms existing integrated solutions.

Author Response:

We admit that our proposed three-dimensional two-wire waveguide-based THz plasmonic ADM circuit exhibits a larger size than some existing integrated solutions (ex. the ones built on photonic crystal and unclad waveguides on the platform of planar silicon wafers (see Table R1)). Nevertheless, it outperforms

them in several other aspects, such as the reconfigurability via 3D integration, the possibility of tuning, as well as the threshold for entering into production. Additionally, the two-wire ADM circuit is not intended to replace the integrated solutions. Instead, it is expected to provide additional opportunities when designing THz communication networks in the future.

The two-wire THz plasmonic ADM circuit demonstrated in this work was constructed by the seamless connection of modular waveguide components, thus allowing its sectional reconfiguration and trivial assembly with other devices without the need for painstaking alignment. In contrast, the silicon-based ADMs built by arranging the composed elements on the platform (silicon wafer) during the fab production can not change their configuration at later stages. The inter-chip connectorization has been proved challenging (if not impossible), thus resulting in difficulties in integrating ADMs and other elements located on discrete chips for THz communications. Therefore, to divert THz signal in networks, different from the two-wire plasmonic ADM circuit whose configuration can be selectively adjusted, one has to replace the whole silicon chip containing both ADM and other elements and then set up its new optical path.

Moreover, the easily accessible modal field of the 3D two-wire waveguide component makes the dynamic tuning of ADM circuit performance highly possible (several strategies have been demonstrated in Supplementary Note 3). To reschedule THz carriers with different frequencies in real time, one can simply apply external stimulus to the circuit. On the contrary, as the silicon platform features a fixed planar structure and tightly confined modal fields, it is challenging to retrofit reported solutions for tunability.

Furthermore, the cost-effective 3D printing techniques make the proposed two-wire ADM circuit with customized functionalities accessible to essentially any research group, thus democratizing the field of device design and fabrication for the upcoming THz communications. In contrast, the preparation of silicon-based integrated optical circuits generally requires the use of costly top-of-the-line fab infrastructure (lithography systems, deep reactive ion etchers, etc.) and high-cost materials (gold, etc.) in a multi-stage fabrication process. Due to the high cost of template preparation, the reported silicon-based ADM circuits have to enter the mass production stage to achieve a competitive unit price for a unified device. This has to await the long-term development and commercialization of terahertz communications currently being studied in the laboratory.

Additionally, due to their unique characteristics, the two-wire plasmonic and silicon-based THz ADM may find applications in different fields in the future. For the one proposed in this work, it could fit for the size-insensitive communication network requiring customizable and reconfigurable configuration, such as signal routing within a data center. While owing to its compact planar structure, the silicon-based ADM is prone to be used in the universal terminal equipment.

Author Action:

A1. Advantages of the proposed two-wire plasmonic circuits over the existing integrated solutions (integrated optical circuits built on the silicon wafer) have been detailed in the second paragraph of Supplementary Note 1.

“In our work, we pursued a combination of 3D printing and metallization techniques as an alternative to CNC machining and microfabrication for the fabrication of THz plasmonic components. In addition to being cost-efficient and requiring relatively cheap infrastructure, this fabrication route allows innovative 3D waveguide and packaging designs, as well as integration of the freeform elements, thus capable of realizing plasmonic components that are far more advanced than classic two-wire or parallel-

plate waveguides [4]. Compared with the silicon wafer-based integrated optical circuit whose template preparation requires costly top-of-the-line fab infrastructures, thus developed two-wire THz plasmonic circuit is easier to enter into production with competitive unit price. Furthermore, terahertz components demonstrated in this work feature inherently modular design with the waveguides, packaging, and alignment elements all integrated into the same component in a single fabrication step. This modular architecture allows trivial assembly and sectional reconfiguration of complex terahertz circuits without the need for painstaking alignment of the individual components. In contrast, to divert THz signal in system containing the existing integrated solutions, one has to replace the whole silicon chip and then set up its new optical path. Moreover, ease of access to the modal fields in the inter-wire air gap opens new ways of dynamic tuning of the optical component performance via various means, which is challenging for the existing silicon wafer-based integrated solutions due to their fixed planar structure and tightly confined modal fields.”

4. Y. Cao, K. Nallappan, H. Guerboukha, G. Xu, and M. Skorobogatiy, “Additive manufacturing of highly reconfigurable plasmonic circuits for terahertz communications,” *Optica* **7**(9), 1112-1125 (2020).

A2. The potential application of silicon-based integrated solutions and two-wire plasmonic ADM circuits in THz communication networks has been discussed at the end of Supplementary Note 1:

“In summary, tandem ADMs that allow multiple THz signals at different carrier frequencies to be selectively dropped, added, or guided through are the key enabling element of the upcoming terahertz communications networks. We believe that our integrated circuits based on 3D printed two-wire waveguides could offer a pathway for the scalable manufacturing of such devices. In the future, unlike existing integrated solutions in a compact and fixed planar structure for universal terminal equipment, the two-wire THz plasmonic ADM circuit can find applications in size-insensitive networks requiring customizable and reconfigurable configuration by taking advantage of its unique characteristics.”

Reviewer 2:

R4 I thank the authors for the new experiments regarding the thermal tunability. However, I regret to say that the experimental results are not convincing. There is no clear change in the minimum position as the temperature increases. I believe the main problem here was the resin which did not have a large enough thermal expansion coefficient. I believe other resins can be chosen to meet that goal. I think the authors should 1) recognize the limit of the thermal tunability of their prototype (in the paper and/or supplementary), and 2) state that alternative resin materials could be used to achieve this goal.

Author Response:

This is indeed a valid point. We agree with the reviewer that the photosensitive resin used in this work (PlasClear 2, Asiga) features a low thermal expansion coefficient and can not be used for thermal tuning. The thermal and mechanical tuning of two-wire ADMs is highly possible when using other kinds of resins in 3D printing.

Author Action:

A1. We recognized the limit of our two-wire ADM prototype for thermal and mechanical tuning and briefly indicated the means to achieve this goal in the sixth paragraph of Chapter 1.

“Moreover, the frequency response of various components developed in this work (Bragg grating, directional coupler, ADM) can be readily tuned. This can be accomplished by dynamically changing the physical properties or the geometry of the plastic substrate and/or a metallic film using either thermal or mechanical activation.”

A2. In addition, we removed Fig. S2 presenting the spectral response of the WBG under different temperatures and revised Supplementary Note 3 to supplement the information mentioned in main text.

“... Considering that the two-wire waveguide-based ADM circuits presented in this work are built on a 3D printed resin substrate, the thermal and mechanical tunability is possible by changing the physical properties and geometry of the resin substrate. Thermal tuning is available by changing the temperature of the frequency-dependent element, i.e., the WBG of an ADM. Furthermore, mechanical tuning of an ADM presented in this work is possible using the grating-loaded side coupler built on a compressible dielectric support. Under axial strain, the deformation would result in the change of a grating period, and as a consequence, a shift in the Add/Drop frequency. Practically, this can be realized by using a variety of commercial photosensitive resins featuring large thermal expansion coefficient (ex. shape-memory polymer) or low Young’s modulus value (ex. Monocure FLEX100 Resin) when 3D printing the ADM structure ...”

R6 I appreciate the new Fig. 2. It conveys the important information in a more concise manner. However, I believe the units should be in dB scale (power) in all figures. The authors use dB values at many places when discussing the text. For cohesiveness, the same should dB scale should be used in all the figures.

Author Response:

We highly appreciate this comment. Indeed, for the convenience of reading, the unified unit of dB for the transmittance and reflectance of proposed two-wire waveguide components should be used throughout both the main text and Supplementary Notes. Therefore, we have revised all the figures,

except Figs. 4(b-e).

When presented in percentage, Figs. 4(b-e) can explicitly show the spectral response of different ports in two grating-loaded side couplers, especially for the Drop and Through ports under In and Add ports launching conditions for channel Drop and Add actions, respectively. In contrast, when converting into dB, due to the considerable difference in the transmittance/reflectance of different ports within the grating stopband, the spectral response difference between idle ports (Through and Add (In and Drop) ports under In (Add) port launching condition) rather than the desired ports (In and Drop (Add and Through) ports under In (Add) launching condition) is emphasized, thus distracting readers' attention. Therefore, we kept the original Fig.4 in the revised main text, and its vital data (transmittance of Drop and Through (Through) ports under In (Add) port launching condition for channel Drop and Through (Add) action) is conveyed in dB in Fig. 5.

Author action:

We revised Figs. 2 and 5 in the main text, as well as Figs. S2, S4, S8, S9, S12, and S13 in Supplementary Notes.

R14 OK. I thank the authors for the new experiments involving a straight waveguide. They are convincing. Small note: in the new text in the paper, “complimentary” should be “complementary”.

Author Response:

Thank you for your comment! We have carefully proofread the main text and supplementary notes to correct errors.

Additional comment: I have been aware of a recently published article in Nature Communications with a topic similar to the work by Cao et al.:

Dong, J., Tomasino, A., Balistreri, G. et al. Versatile metal-wire waveguides for broadband terahertz signal processing and multiplexing. Nat Commun 13, 741 (2022). <https://doi.org/10.1038/s41467-022-27993-7>

This was published on Feb. 8, 2022 (which I assume had a review schedule similar to this work by Cao et al.). I note that in the work by Dong et al., the authors study a wire-based THz waveguide for signal processing and multiplexing. While the topic is similar, there are still, I believe, substantial differences between the 2 works. 1) Dong et al. did not perform any communication experiments and only characterized their system with a time domain spectrometer, 2) Dong et al. proposed a waveguide geometry for polarization division multiplexing, while Cao et al. proposed a waveguide for frequency division multiplexing. Following these observations, I believe there is still room for publication for the manuscript by Cao et al. especially since they were submitted around the same time. In my opinion however, the authors from Cao et al. should at least recognize this paper in their manuscript by properly citing it.

Author Response:

We highly appreciate this comment. Inspired by this excellent work exploring the application of two-wire waveguides in THz polarization-division multiplexing, we believe that multi-channel THz networks employing multiple modulation modalities can be readily realized based on the proposed modular two-wire waveguide platform.

Author Action:

We add the following information in the sixth paragraph of Chapter 1:

“On the component integration level, the methods developed in this work show a pathway to the highly reconfigurable, modular construction of complex terahertz circuits, where individual components can be easily added, replaced, or subtracted to modify the device functionality. In addition to the FDM demonstrated in this work, the developed waveguide circuits can be also used in a four-wire configuration to realize THz Polarization-Division Multiplexing [33], thus paving a way for supporting several modulation modalities in a single device.”

33. J. Dong, A. Tomasino, G. Balistreri, P. You, A. Vorobiov, É. Charette, B. L. Drogoff, M. Chaker, A. Yurtsever, S. Stivala, M. A. Vincenti, C. D. Angelis, D. Kip, J. Azaña, and R. Morandotti, “Versatile metal-wire waveguides for broadband terahertz signal processing and multiplexing,” *Nat. Commun.* 13(1), 1-8 (2022).

REVIEWERS' COMMENTS:

Reviewer #1 (Remarks to the Author):

The achieved device performance is below academic standard. There is no significant scientific novelty in the concept, execution, or device platform. And finally, the claimed benefits of their chosen approach are dubious at best.

No rebuttal will ever change these salient facts, as the issues with the work are fundamental.

This manuscript is not worthy of publication as a journal article in Nature Communications.

Reviewer #2 (Remarks to the Author):

I thank the authors for addressing all my comments.

I believe the new dB-scales convey the performance of their device in a more convincing manner. Indeed, having transmission of 50% may look bad for non-experts, but having insertion losses of 3 dB is totally acceptable in the context of communications, where the signals are generally transmitted with 10s of dB. Similarly, the “jagged frequency response” may look bad on a linear scale but is totally acceptable in a dB scale (again in the context of communications). I believe the authors have made a convincing case by comparing their design with the state-of-the-art in THz and IR. They also pointed to limitations of their design, which can be tackled in future work (it becomes an engineering problem, rather than a purely scientific one).

Also, as I mentioned in my previous review, a recent paper just got published in Nature Communications having a similar design for polarization-division-multiplexing (Dong, J. et al. Versatile metal-wire waveguides for broadband terahertz signal processing and multiplexing. Nat Commun 13, 741 (2022). <https://doi.org/10.1038/s41467-022-27993-7>). In my opinion, frequency-division-multiplexing (as Cao et al. have showed in this paper) is more valuable, as in principle it allows for more multiplexing potential (only 2 possible channels of polarization, compared to >2 possible channels in frequency domain). Given that this paper was recently published in Nature Communications (with a similar review schedule), I believe the paper by Cao et al. has enough similar novelty to be accepted for Nature Communications.

For convenience, we indicate in **bold** the original questions by the reviewer, and in regular font our response to the reviewer.

Reviewer #1:

The achieved device performance is below academic standard. There is no significant scientific novelty in the concept, execution, or device platform. And finally, the claimed benefits of their chosen approach are dubious at best.

No rebuttal will ever change these salient facts, as the issues with the work are fundamental.

This manuscript is not worthy of publication as a journal article in Nature Communications.

Reviewer #2 (Remarks to the Author):

I believe the new dB-scales convey the performance of their device in a more convincing manner. Indeed, having transmission of 50% may look bad for non-experts, but having insertion losses of 3 dB is totally acceptable in the context of communications, where the signals are generally transmitted with 10s of dB. Similarly, the “jagged frequency response” may look bad on a linear scale but is totally acceptable in a dB scale (again in the context of communications). I believe the authors have made a convincing case by comparing their design with the state-of-the-art in THz and IR. They also pointed to limitations of their design, which can be tackled in future work (it becomes an engineering problem, rather than a purely scientific one).

Also, as I mentioned in my previous review, a recent paper just got published in Nature Communications having a similar design for polarization-division-multiplexing (Dong, J. et al. Versatile metal-wire waveguides for broadband terahertz signal processing and multiplexing. Nat Commun 13, 741 (2022). <https://doi.org/10.1038/s41467-022-27993-7>). In my opinion, frequency-division-multiplexing (as Cao et al. have showed in this paper) is more valuable, as in principle it allows for more multiplexing potential (only 2 possible channels of polarization, compared to >2 possible channels in frequency domain). Given that this paper was recently published in Nature Communications (with a similar review schedule), I believe the paper by Cao et al. has enough similar novelty to be accepted for Nature Communications.

Author Response:

We would like to thank the reviewers for their time and thoughtful comments on our work. All the reviewers' concerns have been addressed, which are listed in previous response letters and can be found in the main text and supplementary notes.